# How large does a large ensemble need to be?

Sebastian Milinski[1], Nicola Maher[1], and Dirk Olonscheck[1]

[1]Max Planck Institute for Meteorology, Hamburg, Germany

**Correspondence:** Sebastian Milinski (sebastian.milinski@mpimet.mpg.de)

**Abstract.** Initial-condition large ensembles with ensemble sizes ranging from 30 to 100 members have become a commonly used tool to quantify the forced response and internal variability in various components of the climate system. However, there is no consensus on the ideal or even sufficient ensemble size for a large ensemble. Here, we introduce an objective method to estimate the required ensemble size that can be applied to any given application and demonstrate its use on the examples of global mean near-surface surface air temperature, local temperature and precipitation, and variability in the ENSO region and central United States for the Max Planck Institute Grand Ensemble (MPI-GE). Estimating the required ensemble size is relevant for designing or choosing a large ensemble, but also for designing targeted sensitivity experiments with a model. Where possible, we base our estimate of the required ensemble size on the pre-industrial control simulation, which is available for every model. We show that more ensemble members are needed to quantify variability than the forced response, with the largest ensemble sizes needed to detect changes in internal variability itself. Finally, we highlight that the required ensemble size depends on both the acceptable error to the user and the studied quantity.

## 1 Introduction

Single model initial-condition large ensembles (SMILEs) are a valuable tool to cleanly separate a model's forced response from internal variability and to improve our understanding of the observed trajectory of the climate system in the past, and its projected future evolution (Zelle et al., 2005; Deser et al., 2012a; Rodgers et al., 2015; Kay et al., 2015; Maher et al., 2019; Branstator and Selten, 2009; von Känel et al., 2017; Kirchmeier-Young et al., 2017; Frankignoul et al., 2017; Stolpe et al., 2018).

The ensemble sizes currently available for individual global coupled climate models largely differs. The single-model ensembles within the Coupled Model Intercomparison Project Phase 5 and 6 (CMIP5, CMIP6) are on the low end of available ensemble sizes, typically ranging from three to ten ensemble members for a model, with the majority of models having only one member available. In contrast, computationally expensive SMILEs position themselves on the top end of available ensemble sizes, providing up to 200 ensemble members for a single model and forcing scenario. While studies are beginning to compare multiple SMILEs (Maher et al., 2018; Deser et al., 2019), there is still no clear consensus on how large such an ensemble should be for any given application.

We here introduce a new framework to objectively estimate the required ensemble size for different types of questions and make use of a model's pre-industrial control simulation where possible. Using the pre-industrial control simulation allows us

to estimate the required ensemble size for a specific model even if no large ensemble is available. The objective approach can also help to allocate resources more efficiently (Ferro et al., 2012) and to inform the modelling community how many ensemble members are desirable for CMIP models.

One of the most common applications of SMILEs is to separate a forced response due to anthropogenic global warming from the noise of internal variability. In a sufficiently large ensemble the ensemble mean can be used as an estimator for the forced response (Frankcombe et al., 2018). This approach has been applied to study various regions and quantities.

On a global scale, Deser et al. (2012b) investigate the forced response in temperature and precipitation. They found that around 10 ensemble members are sufficient to detect changes in the global mean land temperature in the next decade, while more than 40 ensemble members are required to detect changes in precipitation. When going further into the future when the signal becomes larger, they find that fewer members are sufficient to detect a forced change. If the signal is large enough, a single ensemble member is sufficient to detect a significant change compared to present-day conditions. This happens when the trajectory of the single member emerges from the range of internal variability for present day conditions.

On both global and regional scales, Olonscheck and Notz (2017) used both the CMIP5 multi-model ensemble and the MPI-GE to conclude that multiple small ensembles from different models are useful to quantify the response uncertainty across different models.

While a forced response in global mean temperature only requires a relatively small ensembles size, forced changes on a smaller regional scale can be more difficult to detect because of the larger variability. Li and Ilyina (2018) investigated the ocean carbon sink and found that up to 79 ensemble members are required to isolate a forced decadal trend in the RCP4.5 scenario in the Southern Ocean, a region with large internal variability. Steinman et al. (2015) quantify the forced response in North Atlantic temperature and argue that for this region, more than four ensemble members are required for a robust estimate of the forced response from a SMILE. Although the objective of the two studies is similar—identifying a forced response—the required ensemble size is very different, indicating that different regions and quantities can have very different requirements on the ensemble size.

In addition to investigating forced changes to anthropogenic forcing, large ensembles also allow an investigation of forced responses to other external forcings such as volcanic eruptions. For regional temperature changes, Pausata et al. (2015) find that up to 40 ensemble members are necessary for a robust detection of a temperature response after a volcanic eruption. Bittner et al. (2016) investigate changes in atmospheric circulation after a volcanic eruption. They analyse the polar vortex and find that the required ensemble size to detect changes in the zonal wind after a strong volcanic eruption depends on the latitude: 7 members are sufficient at the southward flank of the maximum positive wind anomaly, but up to 40 members are necessary to identify a response at high northern latitudes. The ratio of the signal to the noise from internal variability is different in different regions because both the signal but also the internal variability are different. The target of Bittner et al. (2016) was to detect a change in the circulation that is different from zero, but not to quantify it. Quantifying the magnitude of the forced response may require an even larger ensemble size for this application.

Large ensembles have also been used to quantify internal variability, with some studies arguing that very large ensemble sizes are necessary: Daron and Stainforth (2013) conclude that an ensemble with several hundred members is required to

characterise a model's climate, while Drótos et al. (2017) demonstrate that 100 members are sufficient. On the other hand, some studies argue that the pre-industrial control simulation is sufficient to quantify internal variability and no large ensemble is required. Thompson et al. (2015) argue that the pre-industrial control simulation can be used to provide a robust estimate of internal variability and represent future internal variability, implying that a single ensemble member for each model may be sufficient. However, this approach only works if the internal variability does not change over time. In addition, a single realisation for a transient scenario does not allow a clean separation of the forced response and internal variability, even if the magnitude of the internal variability is quantified using a pre-industrial control simulation.

ENSO variability and its potential changes under global warming have been investigated in several studies and widely different future changes have been identified (Stevenson et al., 2012; Bellenger et al., 2013; Christensen et al., 2013). Maher et al. (2018) investigate ENSO variability and its potential changes under global warming in several large ensembles. They find that at least 30 ensemble members are required for a robust estimate of ENSO variability. When using a smaller ensemble, sampling uncertainty may lead to false detection of a forced change in ENSO or a robust difference between two models.

All of the aforementioned studies demonstrate that different applications require different ensemble sizes. However, these studies suffer from two drawbacks. First, the required ensemble size can only be estimated once a signal has been identified in a large ensemble, which requires the large ensemble to exist and be large enough in the first place. Second, the result might be model dependent and may only provide a very rough estimate of the required ensemble size when addressing the same question with a different model.

In this paper, we introduce a basic recipe for estimating the required ensemble size in section 3. The required or ideal ensemble size depends on the region and quantity that is investigated and the type of question. Therefore we differentiate three types of questions that represent questions typically addressed with large ensembles:

1. How many ensemble members are required to identify the response to a change in the external forcing? (Section 4.1)

2. How many ensemble members are required to adequately sample the spectrum of internal variability? (Section 4.2)

3. How many ensemble members are required to identify a forced change in internal variability (e.g., a mode of variability such as ENSO)? (Section 4.3)

An additional discussion of caveats associated with the choice of sampling method is discussed in Appendix A and is relevant for users of the approach proposed in this study.

## 2    Model

In this study, we are using simulations from the Max Planck Institute Grand Ensemble (MPI-GE). The MPI-GE consists of large initial-condition ensembles for several experiments with the Max Planck Institute Earth System Model (MPI-ESM) in its low-resolution configuration. Ensemble members are generated by sampling different years from a 2000-year pre-industrial control simulation for the initial conditions (macro-initialisation). The forcing for the experiments follows the protocol of the

CMIP5 simulations (Taylor et al., 2012). The model configuration and experiments are described in more detail in Maher et al. (2019).

In this study, we use three experiments from the MPI-GE:

- pre-industrial control simulation (2000 years)

- historical simulations (1850-2005, 200 members)

- 1% $CO_2$ simulations (156 years, 100 members)

Note that only the first 100 historical realisations are described in Maher et al. (2019). Realisations 101–200 were added later and use the same configuration as the first 100 realisations, but are initialised from different years of the pre-industrial control simulation.

## 3   A simple method to estimate the required ensemble size

In this section, we use a simple example to design a generic recipe for estimating the required ensemble size for any given application. In the following sections 4.1 to 4.3, we then apply this recipe to various examples.

One of the most common applications of a large ensemble is the separation the forced response and the random internal variability in a time-series. Each realisation from a large ensemble is subject to the same external forcing. Due to different initial conditions, each realisation is a combination of the forced response due to this external forcing and a unique trajectory of quasi-random internal variability. By averaging over a large number of realisations, internal variability cancels out and the forced response remains (Frankcombe et al., 2015). Therefore, the ensemble mean of a large ensemble is often referred to as the forced response. Figure 1 shows the ensemble mean global mean near-surface air temperature (GSAT, blue line) of 200 realisations with CMIP5 historical forcing from the MPI-GE (Maher et al., 2019). Because of the large ensemble size and the use of a globally averaged quantity, the 200-member mean is a clean estimate of the forced response.

Assuming that the 200-member mean provides a good estimate of the forced response, we can then subset the large ensemble to investigate how well the ensemble mean of a smaller ensemble can isolate the forced response. We draw 1000 random samples of sets of 3 members from MPI-GE without replacement. For each of these samples, the 3-member ensemble mean is computed. The red envelope in figure 1 shows the range of these 1000 samples of a 3-member mean forced response. Compared to individual realisations (grey envelope), a 3-member mean reduces internal variability, but can deviate substantially from the 200-member mean. Repeating this analysis for 10, 20, and 50 members shows that a larger ensemble size can separate the forced response from internal variability more effectively.

To quantify how effective the separation of forced response and internal variability is, we show the root-mean-square error (RMSE) of ensemble means for different ensemble sizes compared to the 200-member mean. The solid black line in figure 2 shows how the expected RMSE decreases with increasing ensemble size until reaching zero for 200 members. By choosing an acceptable error, we can then determine the required ensemble size. For example, an acceptable error of 0.02°C would mean

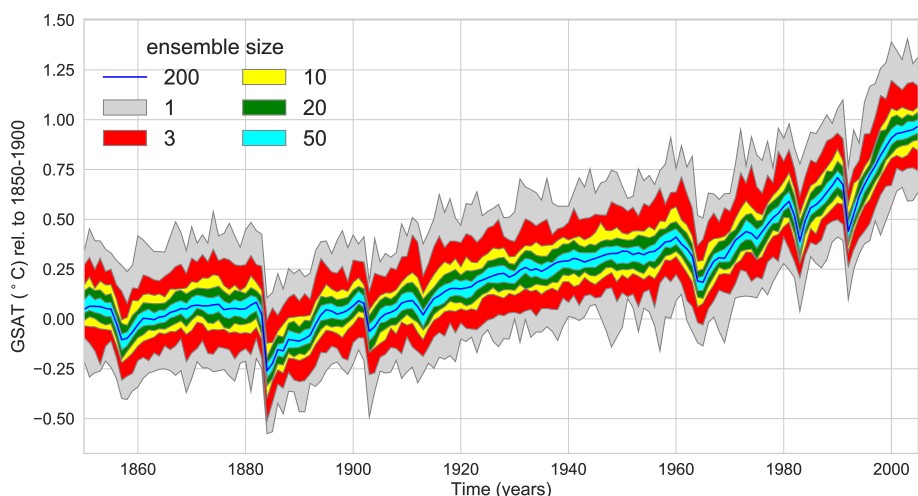

**Figure 1.** The forced response can be quantified using the ensemble mean in a large ensemble, while the ensemble mean of smaller ensembles still contains a contribution from internal variability. The figure is based on global and annual mean near-surface air temperature from the MPI-GE 200 member historical ensemble. The dark blue line shows the 200-member ensemble mean time series. Shaded regions show the range of forced responses estimated by resampling 1000 times for various ensemble sizes. The light grey shading shows the range of the full ensemble, i.e. the minimum to maximum of all 200 realisations for every single year.

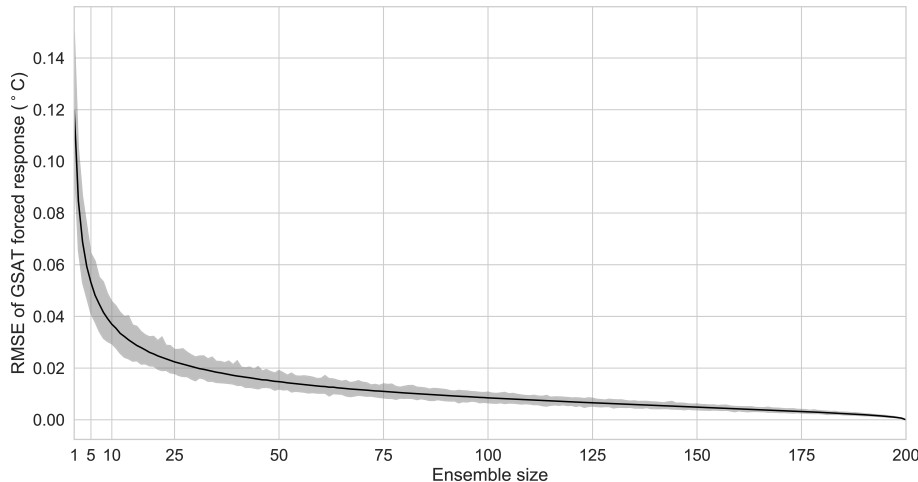

**Figure 2.** A larger ensemble allows a more accurate quantification of the forced response. The black line shows the mean RMSE for GSAT for ensemble sizes from 2 to 200. The reference is the 200-member mean from figure 1 and the RMSE is computed for all 1000 samples. The shaded area shows the range of RMSE values for individual samples, the solid line shows the mean RMSE.

that an ensemble with approximately 50 members is required. We will return to the discussion of what constitutes an acceptable error in the examples in sections 4.1 to 4.3.

While a reduction in the error with increasing ensemble size is expected and indicates that a larger ensemble allows a more accurate representation of the forced response, the vanishing error when using 200 members occurs by construction because we assume that the 200-member mean represents the true forced response. How fast the error is converging therefore depends on how the random samples are generated.

## 3.1 A cautionary note on resampling

One difficulty when determining the required ensemble size for a specific question is the chosen sampling approach: in this study, we generate synthetic ensembles of different ensemble sizes by randomly sampling members from a 200-member ensemble without replacement. Samples generated in this way are not fully independent when approaching the full ensemble size. For example, two random samples of 190 out of the available 200 members will share most of their members. This resampling introduces a problem when the signal is defined by using the full ensemble. Any subsample that is close to the full ensemble size will then indicate that the ensemble size is sufficient by construction.

The resampling problem occurs with any limited sample. At some point, the 1000 random subsamples are not independent anymore because they share many of the randomly drawn members from the full ensemble. Therefore, they look more similar to each other, but also more similar to the 200-member mean. To demonstrate how this resampling affects our estimate of the error, we deliberately reduce the size of the ensemble. For instance, by only using the first 150 members and repeating the analysis. In an empirical analysis, we find that samples using more than 50% of the available ensemble size to generate random samples lead to a substantial bias in the error estimate. We therefore recommend to treat results indicating that e.g. more than 100 out of 200 members are required with caution because the true required ensemble size might be much larger. A more detailed discussion is provided in Appendix A.

## 3.2 A recipe for estimating ensemble size

Based on the example introduced in this section, we suggest the following approach to derive a robust estimate of the required ensemble size for any application. This method can either be applied to one of the existing large ensembles, as shown above for the MPI-GE, or to a long control run, which is available for all models participating in CMIP. We summarise the method in five steps before applying it to several examples in the next section:

1. Define the question to be addressed (isolate a forced response, quantify variability, detect a change in variability).

2. Choose an error metric (e.g. RMSE or variance across samples) and an upper threshold based on the maximum error that is acceptable in the specific application.

3. Estimate the error for different ensemble sizes by subsampling a long control run or a large ensemble of transient simulations.

4. Determine the minimum ensemble size that is required to reduce the error below the threshold chosen in step 2.

5. If the ensemble size determined in this way is less than 50% of the available sample size (e.g. 50 members when subsampling a 100-member ensemble), then the estimated required ensemble size provides a robust estimate for the specific question and model investigated. If the estimated required ensemble size is larger than 50% of the available sample size, then the estimate is biased low and the true required ensemble size could be substantially larger.

## 4   Estimating the required ensemble size: applications

In this section we use the pre-industrial control simulation and transient forced simulations from the MPI-GE to estimate the required ensemble size for a variety of applications, ranging from global to regional quantities. We investigate the different aspects of quantifying the forced response or quantifying internal variability.

### 4.1   Quantifying the forced response

The forced response shown in figure 1 contains various signals. The most prominent signal is the long-term warming trend caused by anthropogenic greenhouse gas emissions. On shorter time scales, volcanic eruptions lead to a cooling of the global mean surface temperature.

In the first example, we continue to use the RMSE to quantify how well the entire forced response is estimated, but we move from the global mean to the regional forced response in near-surface air temperature in the historical runs from the MPI-GE. In figure 3 a–e, the expected RMSE for each grid point is shown for ensemble sizes of 3, 5, 10, 50, and 100 members. This analysis is equivalent to the computation of the mean RMSE for GSAT (black line in figure 2), but applied to each grid point separately. The RMSE is computed as the mean difference between 100 samples and the 200-member mean. When the ensemble mean is based on just 3 members, the expected error in the estimated forced response is large over land regions, in particular in the northern hemisphere. Over the ocean, the RMSE is already small in many regions. Increasing the ensemble size reduces the error. At 50 members, the error is small in most regions of the globe. Because 50 members is smaller than 50% of the maximum ensemble size (200 members), the error estimate for this ensemble size is reliable.

To estimate how many members are sufficient to reduce the error below a critical threshold, we first need to determine what is an acceptable error as outlined in step 2 of the recipe. This choice will depend on the region of interest and the accuracy to which the forced response needs to be quantified. In figure 3 f–j, we show how many members are necessary to estimate the forced response in near-surface air temperature for five acceptable errors that were chosen for illustrative purpose. If the acceptable error (RMSE) is 0.1°C, 10-30 ensemble members are sufficient over the tropical ocean, while more than 50 ensemble members are required over most land regions. Beyond 100 members, the resampling problem inhibits reliable estimates of the sufficient ensemble size. For an acceptable error of 0.25°C, less than 10 members are sufficient over most ocean regions, while more than 50 members are required over high northern latitude land regions. For an acceptable error of 0.5°C, only high-latitude land regions require a large ensemble while the forced response over ocean and land regions at lower latitudes can be estimated with less than 10 members.

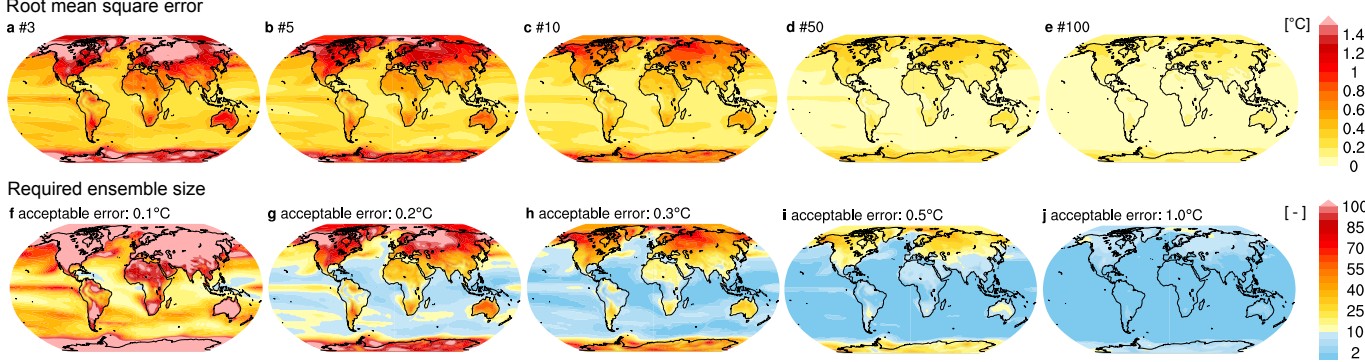

**Figure 3. a-e**, The mean RMSE for the forced response in historical monthly mean near-surface air temperature of MPI-GE for **a**, 3,**b**, 5, **c**, 10, **d**, 50, and **e**, 100 ensemble members relative to the 200-member mean, globally. The RMSE shown here is the mean from 100 random samples without replacement. **f-j**, Required ensemble size to capture the 200-member mean forced response in historical monthly mean near-surface air temperature dependent on the acceptable error of **f**, 0.1, **g**, 0.2, **h**, 0.3, **i**, 0.5, and **j**, 1.0°C.

Conversely for rainfall, the error in estimating the forced signal when using a small ensemble is larger over the tropics than over the higher latitudes (Figure 4 a–e). The largest errors can be found over the Indian Ocean and western tropical Pacific. Similar to temperature, a 50-member ensemble shows very small errors across the globe.

In figure 4 f–j we show how many members are necessary to estimate the forced response with an acceptable error of 0.1, 0.2, 0.3, 0.5, and 1 mm/day. For an acceptable error of 0.2 mm/day, some ocean regions require more than 100 members to capture the forced rainfall response with the required accuracy, while less than 20 members are sufficient over northern Africa and Eurasia. Over large parts of America, between 20 to 40 members are required to estimate the forced rainfall response. For an acceptable error of 0.5 mm/day, 20 to 40 members are required over the Indian Ocean and western tropical Pacific, while less than 10 members are sufficient elsewhere.

For the example in figures 3 and 4, the objective was to isolate the full forced response in a time series, defined as the 200-member ensemble mean time series at every grid point. The full forced response includes all external forcings, both natural and anthropogenic. In many applications, the objective might be to isolate a specific feature of the forced response rather than all components. In the following two examples, we will demonstrate how to estimate the required ensemble size needed to isolate the global warming trend in the 20th century and the global cooling after a major volcanic eruption.

The global warming signal follows a much simpler trajectory than the forced response to all external forcings (cf. figure 1). Here, we fit a linear trend to the historical time series for 1920 to 2005 and define the 200-member mean as the true forced warming trend. Over the 68-year period from 1920 to 2005, the model warms by 0.65 K (figure 5). We acknowledge that a linear trend may not represent the anthropogenic warming accurately, but use this definition to illustrate how a specific aspect of the forced response can be investigated.

We subsample the ensemble for smaller ensemble sizes to generate forced warming trends for smaller ensemble sizes. While the trends in a single realisation can be anywhere in the range from 0.4K to more than 0.8K warming over 68 years, increasing

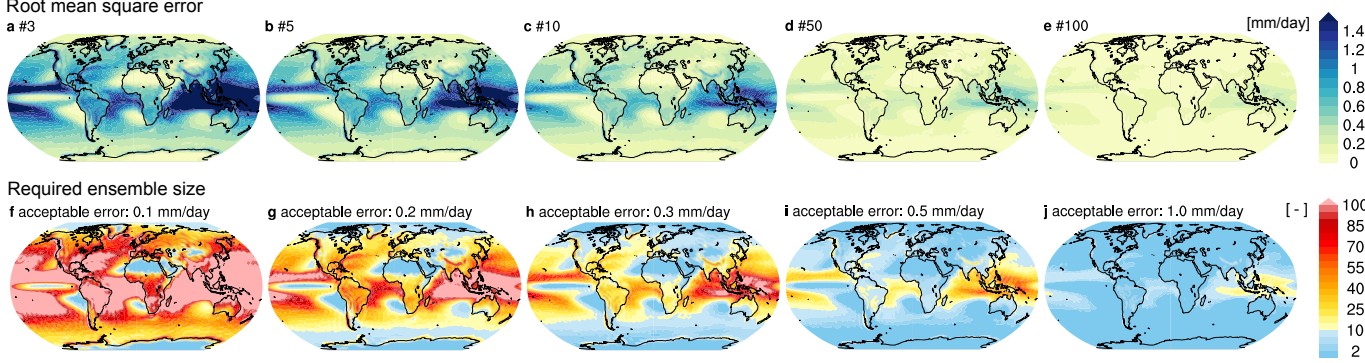

**Figure 4. a-e,** The mean RMSE for the forced response in historical monthly mean total precipitation of MPI-GE for **a**, 3,**b**, 5, **c**, 10, **d**, 50, and **e**, 100 ensemble members relative to the 200-member mean, globally. The RMSE shown here is the mean from 100 random samples without replacement. **f-j,** Required ensemble size to capture the 200-member mean forced response in historical monthly mean total precipitation dependent on the acceptable error of **f**, 0.1, **g**, 0.2, **h**, 0.3, **i**, 0.5, and **j**, 1.0 mm day$^{-1}$.

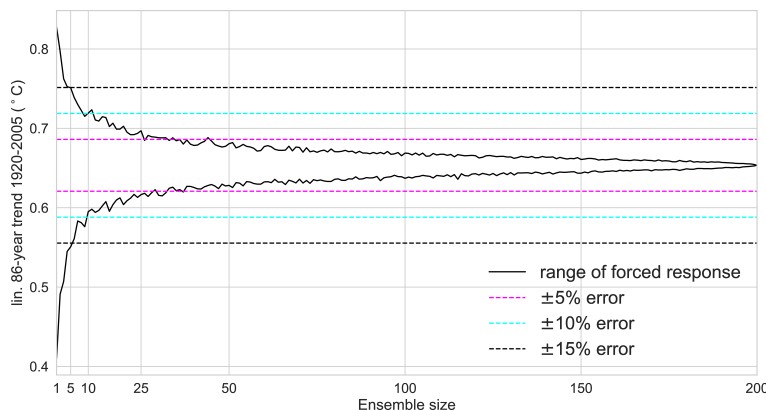

**Figure 5.** Linear warming trend from 1920 to 2005 for different ensemble sizes shown as a linear trend fitted to the ensemble mean. Black lines show maximum and minimum 86-year ensemble mean temperature trend from 1000 random samples. Errors are shown as percentage of the 200-member ensemble mean temperature trend.

the ensemble size to 5 members already leads to a significant reduction in the error (figure 5). The warming trend in every 10-member ensemble is within the 20%-range ($\pm10\%$, cyan dashed lines) of the true warming trend, indicating that ensembles with 5-10 members can provide a good estimate of the forced linear warming trend. While an error within the 20%-range of the true signal may be sufficient for some applications, the acceptable error for other applications might be larger or smaller and result in a smaller or larger acceptable ensemble size. For an acceptable error of $\pm15\%$, 5 ensemble members would be sufficient while for an acceptable error of $\pm5\%$ at least 25 ensemble members are required. All of these error estimates are below 100 members and therefore not dominated by the resampling problem.

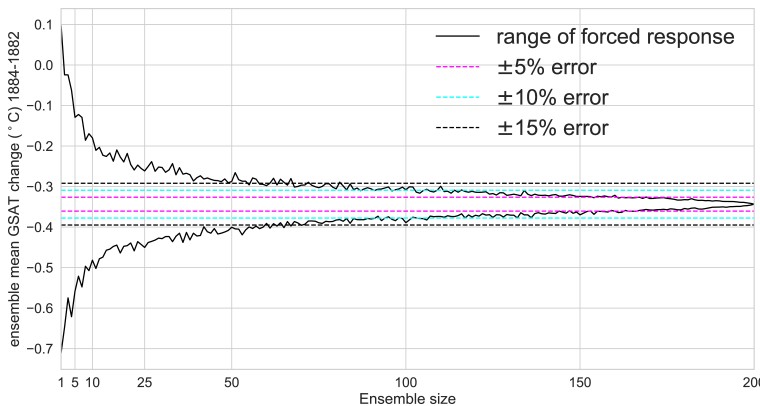

**Figure 6.** GSAT cooling after Krakatoa eruption for different ensemble sizes shown as the ensemble mean temperature difference between 1882 and 1884. Black lines show maximum and minimum temperature response from 1000 random samples. Errors are shown as percentage of the 200-member ensemble mean temperature response.

For signals on shorter time-scales, the required ensemble size can be quite different. In figure 6 we analyse the GSAT cooling after the Krakatoa eruption in 1883. The forced cooling is quantified as the difference between 1884, the year after the eruption, and 1882, the year before the eruption. The 200-member mean shows a forced cooling of -0.34K after the eruption. Due to internal variability, a single realisation can even show a warming after the volcanic eruption. More than 1 member is required

for the ensemble mean to capture a cooling in all samples. However, the ensemble mean cooling for 5 members can still exceed the range from -0.2K to -0.5K. More than 50 ensemble members are necessary to estimate the forced cooling within ±15% of the true forced cooling, and approximately 100 members are required to reduce the error below ±10%. Due to the resampling problem, we cannot derive a robust estimate for the ensemble size required to reduce the error to less than ±5%. While the analysis in figure 6 suggests that 150 members would be sufficient for a ±5% error, this number is close to the full ensemble

size of 200 members and therefore biased low. The true required ensemble size to reduce the error to ±5% is likely larger than 150 members.

These examples demonstrate that the required sample size to estimate the forced response depends on the region and variable (figures 3 and 4), as well as the feature of interest in the forced response (figures 5 and 6). Whereas for some applications 5 members are sufficient to reduce the error to an acceptable magnitude, other applications require at least 50 members. A robust

estimate for the forced response is given by the ensemble mean when averaging over the ensemble attenuates internal variability sufficiently (Frankcombe et al., 2018). The number of members required for this depends both on the magnitude of the forced signal and the magnitude of internal variability, but also on the acceptable error for a specific application.

## 4.2 Quantifying internal variability

While quantifying the forced response only requires a robust estimate of the mean, quantifying internal variability requires

more members because higher order moments of the distribution need to be estimated. In the following two examples, we use

the second statistical moment of the distribution, the standard deviation, to quantify internal variability. We note that if the distribution deviates from a normal distribution, only using the standard deviation to quantify internal variability may not be sufficient.

Here, we investigate internal variability in two regions. The tropical Pacific, where the variability is primarily driven by the El-Niño Southern Oscillation (ENSO), and the central United States (34°N-46°N, 116°W-96°W). The tropical Pacific region shows substantial variability on interannual to decadal time scales. Previous work has demonstrated that large sample sizes are necessary to quantify ENSO variability (Maher et al., 2018; Wittenberg, 2009). As a second region, we analyse temperature variability over the central United States. We hypothesise that these two regions should have different requirements for the ensemble size, with a smaller required ensemble size for the central United States than the tropical Pacific to stay within an acceptable error range.

For the following examples we use the 2000-year pre-industrial control simulation from the MPI-GE. The advantage of this approach, in contrast to the examples for the forced response, is that the required ensemble size can be estimated for any model without needing a large ensemble to be available. The disadvantage is that when using the pre-industrial control simulation, we assume that internal variability does not change under global warming.

We quantify ENSO variability by using the December, January, February (DJF) variability in the Niño3.4 box (5°N-5°S, 170°W-120°W). To ensure that ENSO variability on interannual to multi-decadal time scales is sampled, we use the Niño3.4 standard deviation for a 100-year period. The standard deviation, as computed for the full 2000-year time series is used as the truth in this context and indicated by the horizontal black line in figure 7a. To generate synthetic ensemble members, we split the pre-industrial control simulation into overlapping 100-year segments. Each segment is used as one ensemble member and the temporal standard deviation over the 100-year segment represents ENSO variability for this member. For an ensemble size of one, the spread in ENSO variability seen in figure 7a indicates that individual 100-year periods can have substantially more or less variability than the reference value based on the full control run.

To account for this centennial modulation of ENSO variability, the ENSO variability in multiple ensemble members can be averaged to get a more accurate estimate of the average ENSO variability. We simulate different ensemble sizes by averaging over randomly chosen members for a given ensemble size and repeat this 1000 times. By using a 5-member mean, the error of the estimated variability in all samples is within ±15% of the true value. To reduce the error below ±10%, 10 ensemble members are sufficient. To improve the accuracy so that the ENSO variability estimate is within ±5% of the truth, nearly 50 ensemble members are necessary.

For a region with less variability, much smaller ensemble sizes are sufficient to obtain a similar accuracy. For annual mean central US temperatures (figure 7b) any individual realisation is within ±15% of the truth and 10 members are sufficient to increase the accuracy to the ±5% range around the truth, whereas 50 members are necessary for ENSO. This emphasises that for some regions and quantities, a moderate ensemble size or even a single realisation can be sufficient to quantify internal variability.

In both examples, the long sampling period of 100 years increases the sample size and thereby improves the accuracy for individual realisations. This is useful if the objective is to quantify variability when stationarity can be assumed, but can

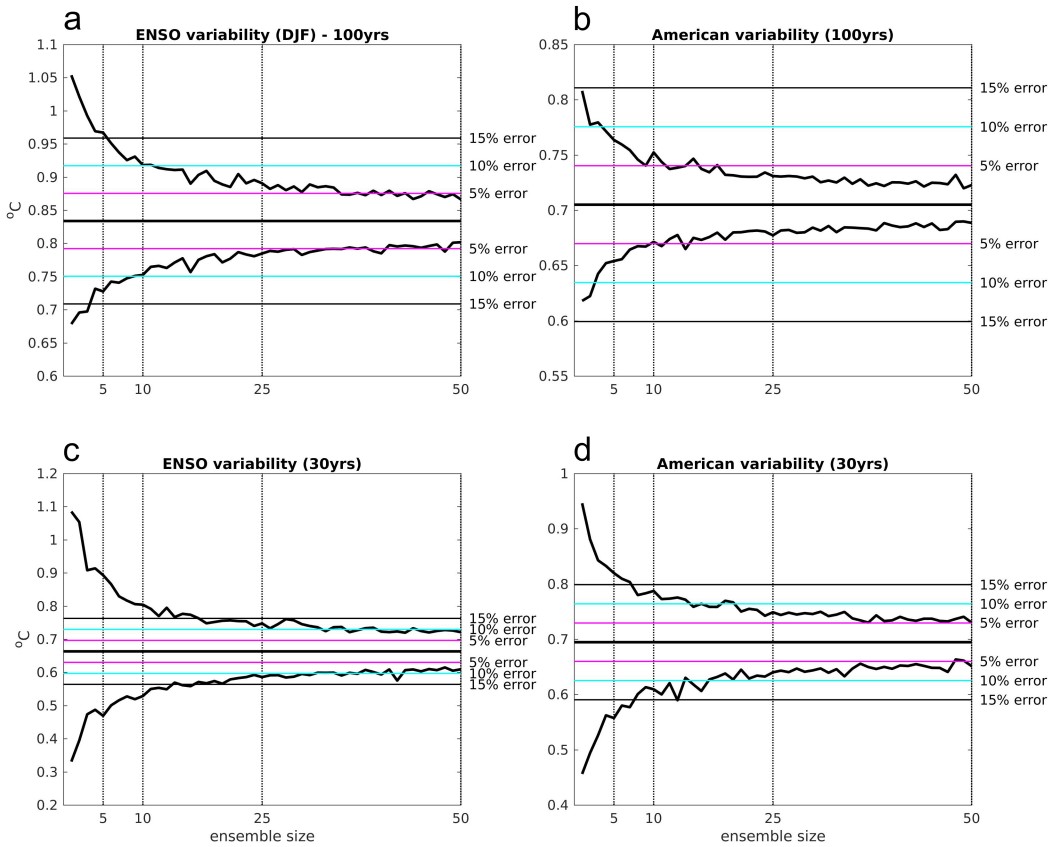

**Figure 7.** We show for increasing ensemble sizes the: a) ENSO variability in the Niño3.4 box (5°N-5°S, 170°W-120°W) calculated over 100 year periods, b) Central United States variability (34°N-46°N, 116°W-96°W) calculated over 100 year periods, c) ENSO variability in the Niño3,4 box calculated over 30 year periods, d) Central United States variability calculated over 30 year periods. All indices are calculated from the 2000 year MPI-GE control run. Each index is calculated as a running value at each time-step in the control. ENSO indices are calculated for DJF and United States indices are calculated for the annual mean. Ensembles of 1 to 120 members are created by randomly sampling the control simulation without replacement. For each ensemble size we create 1000 artificial ensembles. The estimated true value is calculated by using the entire 2000 years of the control and is shown in the horizontal black line. The maximum and minimum values of each index from the 1000 samples are shown in the solid black lines. Varying error thresholds are shown in the horizontal coloured lines.

be problematic if the objective is to identify a change in variability, such as changes in ENSO characteristics under global warming. A more detailed discussion of estimating ENSO variability, in particular using the ensemble dimension instead of the time dimension in transient simulations to quantify internal variability, can be found in Maher et al. (2018) and Haszpra et al. (2020).

### 4.2.1 Notes on sampling from a pre-industrial control simulation

Sampling from a pre-industrial control simulation to estimate the required ensemble size has two advantages: this can be done before producing a large ensemble for the model and is based on a simulation that is available for every climate model in CMIP5 and CMIP6. Different approaches can be used when sampling from a pre-industrial control simulation. In the following, we discuss different options and their advantages and disadvantages.

- *Overlapping segments (applied here):* We choose to use continuous 100-year and 30-year segments to keep temporal autocorrelation intact. From the 2000-year simulation, we can thus generate 20 independent, non-overlapping synthetic realisations (for 100-year segments). To increase the sample size, we allow overlapping segments. These samples are not independent, which leads to a biased estimate as discussed in appendix A, but enables estimates for ensemble sizes larger than 20.

- *Non-overlapping segments:* The advantage of this approach is that synthetic members can be assumed to be independent and temporal autocorrelation is kept intact. However, for long segments or a short pre-industrial control simulation, only a small number of synthetic members can be generated.

- *random year selection to generate synthetic segments or members:* The synthetic segments generated by random year selection allow for a wider variety of samples in a segment than continuous segments sampled from the pre-industrial control simulation. However, information about temporal autocorrelation is lost and synthetic segments could have larger variability than continuous segments in the presence of strong variability on time scales longer than the segment. If the time scale of variability is not the focus of a study, sampling random years to generate synthetic ensemble members can be informative to estimate how well statistics computed across ensemble members (e.g. Maher et al., 2018; Haszpra et al., 2020) capture the model characteristics.

## 4.3 Quantifying changes in internal variability

To quantify changes in internal variability, we need a robust estimate of internal variability both for a reference period and for a period where we want to investigate a potential change in variability (e.g. a pre-industrial control state and a time period in a future scenario). This problem is more challenging than the previous examples because the errors for the variability estimates of the two time periods add up. To demonstrate this, we use the internal variability of September Arctic sea ice area as an example. Previous work has shown that the internal variability in Arctic sea ice area first increases under warming, before it approaches zero when most of the Arctic sea ice has melted (Goosse et al., 2009; Olonscheck and Notz, 2017). We analyse the 100 members from the 1% $CO_2$ scenario from the MPI-GE and use the ensemble standard deviation as an estimator of internal variability. After 120 years, nearly all ensemble members show a completely ice-free Arctic in September (figure B1a). The internal variability increases from model year 1 to year 80, before it sharply drops reaching zero around year 120 (figure B1b).

Here we focus on the increase in variability from the beginning of the simulation to year 80 and ask how many ensemble members are necessary to robustly quantify this change in internal variability. To increase the sample size, we use a decadal

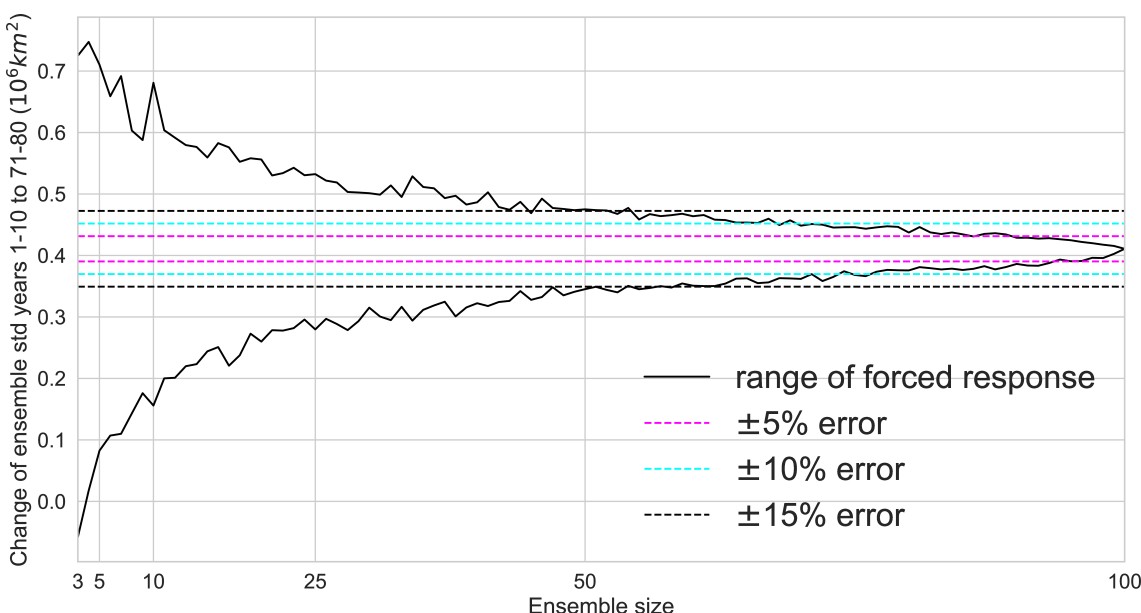

**Figure 8.** Change in internal variability of September Arctic sea ice are from the first decade to years 71–80 in a 1% $CO_2$ experiment. For different ensemble sizes, we compute the ensemble standard deviation and then average for the first decade and years 71–80 before computing the difference. Black lines show maximum and minimum change in variability from 1000 random samples. Errors are shown as percentage of the 100-member variability change.

mean of the ensemble standard deviation rather than a single year. We then compute the difference in internal variability between the two time periods for ensemble sizes between 3 and 100 members. Figure 8 shows the range of this change in internal variability from 1000 random samples. To quantify the change in variability within ±15% of the true value (here defined as the internal variability change estimated with 100 members), 50 ensemble members are necessary. An error of

less than ±10% and ±5% is only reached beyond 50 members. Due to the effect of resampling beyond 50 members, we cannot estimate the required ensemble size for these error thresholds from the 100-member ensemble used here. For very small ensemble sizes, the estimate of the variability change may even show the opposite sign of the true change, i.e. a decrease in internal variability.

    The large number of ensemble members required to robustly quantify this change in variability shows that identifying a

change in internal variability requires the largest ensemble size of all examples shown in this study, even when using decadal averaging to increase the sample size. This is because a robust estimate of a change in internal variability requires a clean separation of internal variability from the forced response and a robust estimate of internal variability for two different time periods. Errors in any of these estimates will propagate to the estimated change in variability, thereby making it more challenging. A small forced change in internal variability will further complicate this analysis.

A first estimate for the magnitude of a detectable change in internal variability can be derived from the control run (as in figure 7). Any change in variability that is smaller than the uncertainty of the estimated internal variability for a given ensemble

size is not detectable. We note that this method can also be used to add error bars to estimates of forced changes in internal variability under climate change in small ensembles or single realisations from CMIP and hence determine the robustness of results.

## 5  Summary and conclusions

Multiple ensemble members for a single climate model are required for robustly estimating the model's forced response to an external forcing change and its internal variability. Without a robust characterisation of these model characteristics, differences between models or a model and observations can easily be misinterpreted as significant differences, while they could be simply caused by an insufficient sample size. Therefore it is important to use an ensemble size that is sufficiently large to allow a robust quantification of the model characteristic that is investigated.

Here we present a generalised approach to estimate the ensemble size that is required to robustly estimate a model's characteristics. While the focus of this study is on the generalised method, the example applications can provide some insight into the required ensemble size for a variety of applications in the MPI-GE. We differentiate three types of question: identifying a forced response, quantifying internal variability, and identifying a change in internal variability. In a next step, an adequate error metric for quantifying the deviations from the true model characteristics is defined and an acceptable error suitable for the application is chosen. By subsampling a pre-industrial control simulation or a large ensemble of transient simulations, the error for different ensemble sizes can be estimated. By applying the previously selected acceptable error as a threshold to these error estimates for different ensemble sizes, the minimum required ensemble size for the given question and model can be determined. Because the subsampling of the full sample does not generate independent samples when approaching the full ensemble size, the error estimate is biased for ensemble sizes close to the available ensemble size. We demonstrate that this resampling effect substantially affects the error estimate when using more than 50% of the full ensemble. For example, a 50 member ensemble cannot be used to conclude that 50 members are sufficient for a given application, because all ensemble estimates beyond 25 members would be affected by resampling and therefore biased.

We apply the method to several examples and use the 200-member historical ensemble, a 2000-year pre-industrial control simulation, and a 100-member 1% $CO_2$ experiment from the MPI-GE to estimate required ensemble sizes for various applications for the MPI-ESM model.

To identify the externally forced temperature response from 1850–2005, most ocean regions require less than 10 members, while land regions at higher latitudes may require more than 50 members. To characterise rainfall changes over the same period, more ensemble members are required in the tropics than in higher latitudes. While regions that require more ensemble members can be objectively identified, the required number of members depends on a subjective choice of the acceptable error and can therefore vary substantially for different applications.

The analysis of the forced cooling after a volcanic eruption and the analysis of ENSO variability demonstrate that a small ensemble size can lead to a misinterpretation. For the example of the volcanic eruption, an ensemble consisting of 2-3 members could show a warming after the volcanic eruption, while the true forced response of the model is a cooling. For ENSO, a too

small ensemble still contains a large uncertainty in the estimate of ENSO variability. This may lead to a misinterpretation of a signal as a forced change in ENSO, whereas it might still be within sampling uncertainty. Wittenberg (2009) show that samples from different time periods in a pre-industrial control simulation can show substantially different ENSO characteristics. Cai et al. (2018) on the other hand use single realisations for different models to identify forced changes in ENSO in future projections. While the robustness of the results seems clear given most models show an increase in ENSO amplitude, we show that within a single model differences between realisations can be large due to internal variability alone. By using the method introduced in this study, we can add to the robustness of studies such as Cai et al. (2018) by adding error bars from the pre-industrial control simulation to each model to test if changes in variability are indeed robust within each model.

The examples in this study demonstrate that for some applications ensemble sizes around 5 members are sufficient while other applications require ensemble sizes well above 100 members. In section 1 we introduced several estimates for required ensemble sizes from the literature. While most of the applications from previous studies are not directly comparable to the examples we use here, the large range of required ensemble sizes emphasizes the need to systematically estimate the required ensemble size for each individual application. Furthermore, the required ensemble size may be model dependent. Therefore, the numbers derived in this and previous studies should only be used as approximate estimates and supported by a systematic model- and application-specific estimate following the approach outlined in this study.

Information about the sufficient ensemble size is not only crucial when choosing or designing a large ensemble, but can also help to identify applications where a small number of ensemble members is sufficient and thereby inform the design of multi-model intercomparison studies. The method introduced in this study can add to the robustness of results both from single model large ensembles and multi-model ensembles.

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

*Author contributions.* All authors conceptualised the study, and carried out the formal analysis. SM wrote the original draft with input from all authors.

*Competing interests.* The authors declare that they have no conflict of interest.

*Acknowledgements.* We thank Chao Li for conducting an internal review of the manuscript, Jin-Song von Storch for helpful comments, and the Max Planck Society for the Advancement of Science for funding all three authors. We thank the two anonymous reviewers for constructive feedback. We thank Mikhail Dobrynin and Johanna Baehr from the University of Hamburg for completing the second hundred MPI-GE ensemble simulations and providing the data from these simulations for use in this paper. DO was supported by the European Union's Horizon 2020 Research and Innovation Programme under grant agreement number 820829 (CONSTRAIN).

## Appendix A: Notes on sampling

In this study, we made several choices on how we sample from a large ensemble or pre-industrial control simulation. In this section, we discuss alternative sampling approaches and caveats.

### A1  Resampling with and without replacement

We choose to resample without replacement for all examples shown. While this choice leads to ambiguities in error convergence
as discussed in the following section A2, we argue that sampling without replacement is a better proxy for what we try to imitate by resampling: a random set of members that we could have produced when running a given number of realisations. Sampling with replacement would mean that for example a randomly sampled 5-member ensemble could contain two (or more) identical

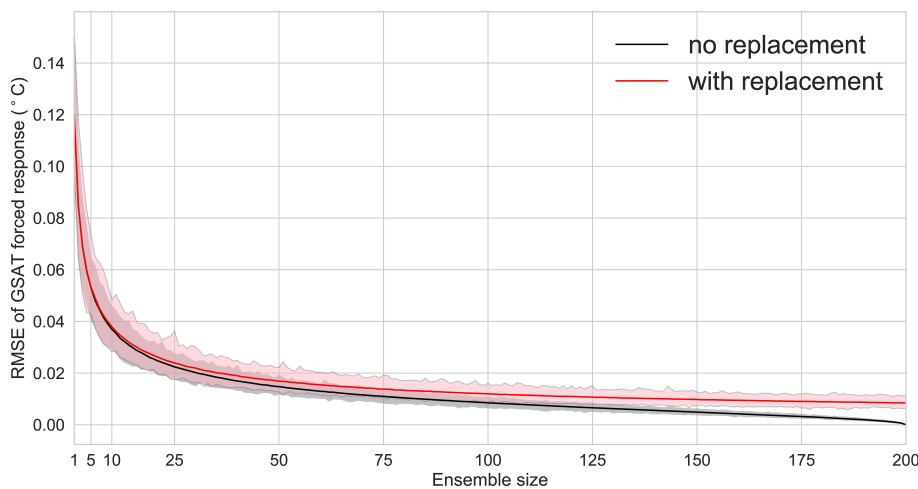

**Figure A1.** Sampling with or without replacement affects the error estimate and therefore the estimate for the required ensemble size. The black line shows the mean RMSE for GSAT for ensemble sizes from 2 to 200. The reference is the 200-member mean from figure 1 and the RMSE is computed for all 1000 samples. The shaded area shows the range of RMSE values for individual samples, the solid line shows the mean RMSE. The red line and shading show the RMSE for ensemble sizes from 2 to 200, but samples are generated by allowing sampling with replacement.

realisations. Given how SMILEs are initialised, this is unlikely to happen and even if it would happen, such an ensemble would not be used as a set of independent realisations without careful investigation.

In figure A1, we repeat the analysis shown in figure 2 but allow replacement when resampling from the 200 members. We still use the 200-member mean as the reference for the forced response in historical GSAT. Sampling with replacement results

5   in a consistently larger error estimate for the mean RMSE, resulting in a larger required ensemble size for a given error.

## A2 How resampling from a small ensemble can bias the error estimate

Generating samples without replacement as applied in this study can bias the error estimate when approaching the full ensemble size. We use the distribution parameters of the full ensemble, e.g. the mean or standard deviation, as the 'truth' in many of the examples shown here. When the size of the sample approaches the size of the full ensemble, for example 190 members from a
200-member ensemble, the difference between these ensembles will be small because they share most of their members. This results in a small error estimate, but does not necessarily mean that 190 members are sufficient for a given application.

The resampling problem occurs with any limited sample. At some point, the 1000 random subsamples are not independent anymore because they share many of the randomly drawn members from the full ensemble. Therefore, they look more similar to each other, but also more similar to the 200-member mean. To demonstrate how this resampling affects our estimate of the
error, we deliberately reduce the size of the ensemble. For instance, by only using the first 150 members and repeating the analysis (purple line in figure A2), the random samples are subsets of these 150 members. Because the 150-member mean is now used as the best estimate, the RMSE is—by construction—zero at 150 members. Similar behavior can be seen when only using the first 100 (red), 75 (green), 50 (blue), and first 20 members (yellow line).

We investigate at which sample sizes the reduction of the error mainly occurs because of an increased ensemble size, or
simply because of resampling that leads to an error convergence without additional information about a sufficient ensemble size. For a smaller number of realisations in the full ensemble, the resampling starts to dominate the error convergence earlier than in a much larger ensemble. Therefore, the comparison of the different maximum ensemble sizes in figure A2 indicates when the resampling begins to affect the error convergence. For ensemble sizes that are much smaller than the maximum ensemble size, the different random samples are largely independent and therefore hardly affected by resampling. When increasing the
ensemble size in the subsamples, the resampling starts to affect the error estimate for a small maximum ensemble size (e.g. 20 members) whereas the samples are still independent when drawn from a much larger maximum ensemble size (e.g. 200 members). The sample size for which the RMSE estimate in a smaller maximum ensemble size starts to diverge from the RMSE estimate based on a larger maximum ensemble size determines the threshold of where resampling substantially affects the error convergence. Beyond this sample size, the error estimate should not be used to approximate the true error.

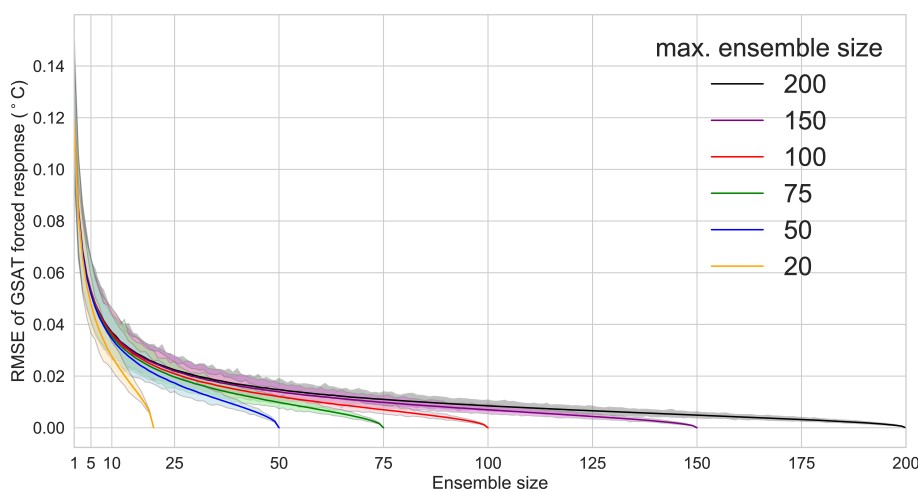

**Figure A2.** In a smaller ensemble, the RMSE converges to zero earlier. This is caused by resampling and does not indicate that the error is small. The black line shows the mean RMSE for GSAT for ensemble sizes from 2 to 200. The reference is the 200-member mean from figure 1 and the RMSE is computed for all 1000 samples. The shaded area shows the range of RMSE values for individual samples, the solid line shows the mean RMSE. The other colors show the same analysis after excluding the last 50 members (purple), 100 members (red), 125 members (green), 150 members (blue), and 180 members (yellow) from the ensemble.

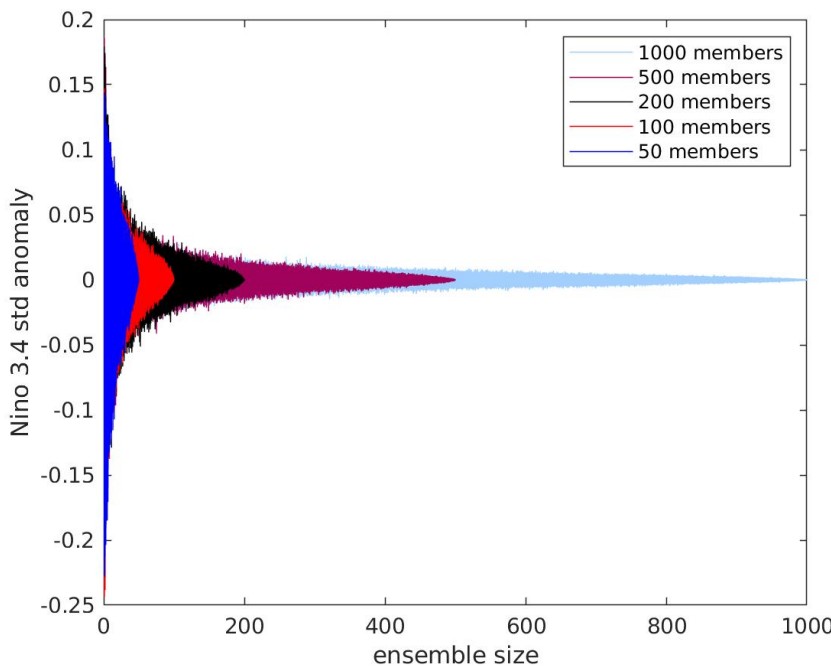

**Figure A3.** PDF of ensemble-averaged Niño3.4 standard deviations possible in the MPI-GE pre-industrial control simulation for subsampling ensembles ranging from 50 to 1000 members (shown as different colors) for smaller ensemble sizes. Each PDF is shown relative to the corresponding ensemble mean value. We use the last 1000 years of the 2000 year control run to calculate the ranges. The Niño3.4 standard deviation is calculated over 50 year periods. The PDFs are created by resampling the control simulation 1000 times. For each PDF the entirety of the 1000 years are used (i.e. the blue 500 member pdf is the mean of 2 500 members PDFs).

We find that the RMSE estimates for different maximum ensemble sizes in figure A2 always start to diverge when about 50% of the maximum ensemble size are used. This implies that up to 50% of the maximum ensemble size can be used to estimate the forced response of GSAT in a transient forcing scenario without a major impact from resampling.

The same resampling problem also occurs for other questions. To demonstrate this, we investigate how many members are
5    necessary to sample ENSO variability. We use the 50-year standard deviation of the Niño3.4 box to quantify ENSO variability. A single 50-year period is treated as one ensemble member. Random subsamples of 50-year periods from the 2000-year pre-industrial control simulation from the MPI-GE are used to generate a synthetic ensemble. In figure A3, the light blue envelope shows that by averaging the standard deviation from more members, a more accurate estimate of ENSO variability can be obtained.
10   We then reduce the maximum ensemble size by using only 500 (200, 100, and 50) years from the control run. Similar to the result in figure A2, the error appears to converge when approaching the maximum ensemble size. By comparing the different

maximum ensemble sizes in figure A3, we can see that the resampling begins to affect the error estimate when the ensemble size approaches 50% of the maximum ensemble size.

These two independent lines of evidence demonstrate that resampling affects the error estimate when using more than 50% of the available maximum sample size (either ensemble members or years in a pre-industrial control simulation). Beyond this

ensemble size, the analysis does not provide a realistic estimate of the error and conclusions about the required ensemble size will be biased low. We note that for very simple applications, such as the mean of a stationary time series, the error scales with $\frac{1}{\sqrt{n}}$. For more complex error estimates, such as the RMSE between non-stationary time series, the scaling law is not as simple, which is why we rely on the empirical analysis outlined above.

## Appendix B: Arctic sea ice area under strong warming

The internal variability of September Arctic sea ice area is known to change under global warming. In this study, we use September Arctic sea ice area as an example for a quantitiy with a change in internal variability under global warming.

Previous work has shown that the internal variability in Arctic sea ice area first increases under warming, before it approaches zero when most of the Arctic sea ice has melted (Goosse et al., 2009; Olonscheck and Notz, 2017). We analyse the 100 members from the 1% $CO_2$ scenario from the MPI-GE and use the ensemble standard deviation as an estimator of internal variability.

After 120 years, nearly all ensemble members show a completely ice-free Arctic in September (figure B1a). The internal variability increases from model year 1 to year 80, before it sharply drops reaching zero around year 120 when all sea ice is lost (figure B1b).

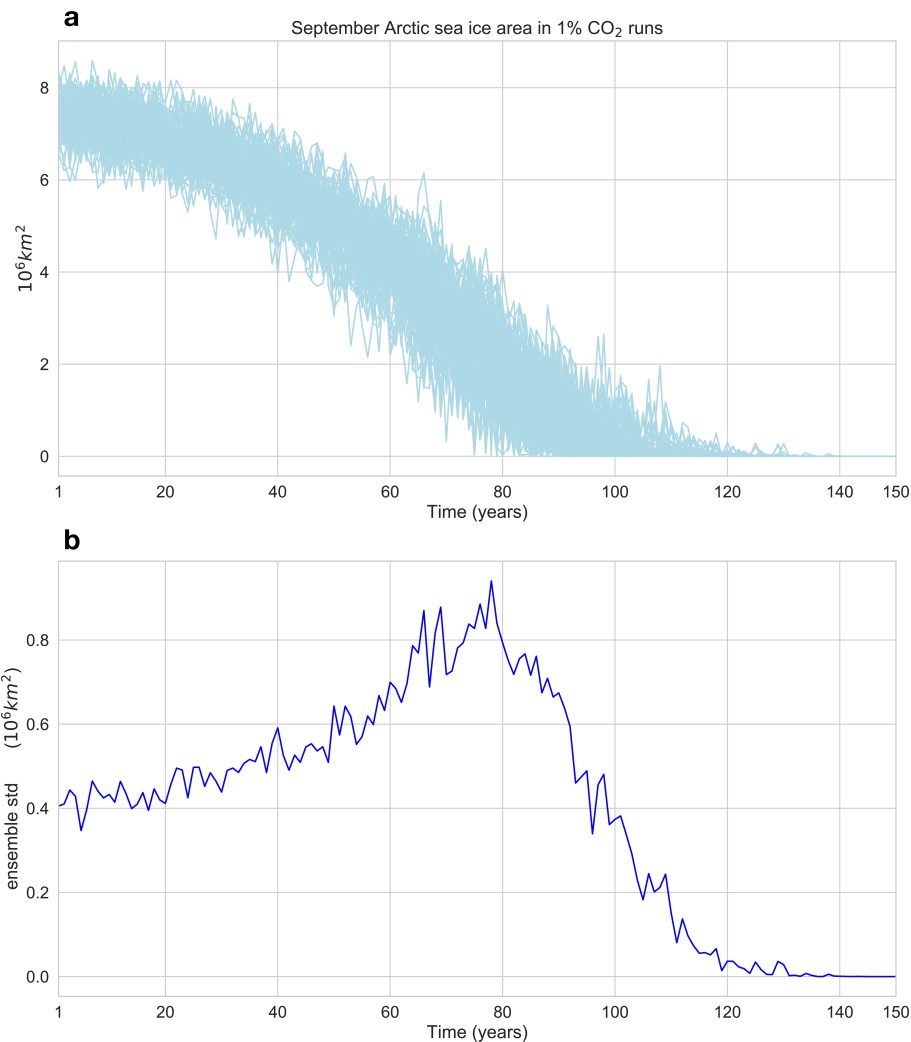

**Figure B1.** a) September Arctic sea ice area in the 100 realisations for the 1% $CO_2$ experiment. b) ensemble standard deviation for the 100 realisations.