# Peer review of "How large does a large ensemble need to be?"

_Earth System Dynamics, 2019_

## Referee Comment (RC1) · Anonymous Referee #1 · 22 Jan 2020

General comments:

In this paper, the authors study the impact of ensemble size on the estimation of different climate statistics using the MPI Grand Ensemble and a pre-industrial control simulation. They analyze the statistical error associated with different quantities as estimated from ensembles of varying sizes, such as the forced response in global surface air temperature, as well as in regional temperature and precipitation. They also assessed the required ensemble size for estimating ENSO variability, linear warming/cooling trends, and changes in internal variability for Arctic sea ice.

Overall, I think this study is highly relevant for guiding users on required ensemble sizes related to different applications, as well as to provide useful insights to climate modellers in the context of the production of upcoming large ensembles. The paper is generally well written and results are original, interesting and worth publishing. How-

ever, there are a few sections that would need to be revisited. For instance, I think a short additional section providing a basic description of the "Data and Methods" would make the paper much easier to understand. In addition, I have some concerns about the selected methods, whose details and implications should be discussed in more details. Finally, the conclusions should better put the original findings into a wider context, especially by comparing with other existing studies (as cited in the introduction) that also have estimated required ensemble sizes.

My main concern about the methodology used in this paper is the exaggerated importance of what the authors call the "resampling problem" (RP). If the aim of this paper is to provide robust estimates of the required ensemble size for different applications (as stated several times in the paper), the importance given to the RP is an obstacle to this goal. The RP is actually an artifact of the selected strategy of resampling the large ensemble without replacement and has profound impacts on the interpretation of the results. With this approach, the question of "How large does a large ensemble need to be?" becomes highly conditional to the size of the ensemble at hand, especially when 50% (here loosely estimated) of the maximum ensemble size is exceeded. If the author would replace their strategy by resampling WITH replacement, the RP would also become a limitation at some point, but for much larger sample sizes (probably even above than the actual maximum ensemble size of 200 members).

The previous comment mainly applies to the results based on MPI-GE, but the issue of the resampling strategy also applies to the results based on the pre-industrial control simulation. For this part, the authors do the resampling by generating synthetic members obtained by splitting the pre-industrial control into overlapping segments (e.g. 50 or 100 years). However, three resampling strategies were actually possible, without any explicit mention in the document: 1) overlapping segments (suffering from the serial dependence of the windows), 2) non-overlapping segments (leading to only 20 members from the 2000-year time series), and 3) random year selection to generate synthetic segments (either with or without replacement). Implications and interpreta-

tion of these possible approaches should be discussed in order to support the decision of selecting which one is better to apply in which context.

Specific comments:

1. p1l7-8 "First, we determine how much of an available ensemble size is interpretable without a substantial impact of resampling ensemble members" The RP is a limitation of the current approach and could be attenuated by changing the resampling approach. I don't think this issue should be mentioned in the abstract, and other similar comments in the paper should be revisited according to the above general comment on RP.

2. P2L13: "to to"

3. P2L22-24: I think the reference to Pausata et al. (2015) is not correct. Maybe another paper from the same author is cited ?

4. P1L24 "make use of a model's pre-industrial control run where possible." This is not that clear in the paper why sometimes we use MPI-GE and otherwise the preindustrial run. This should be clarified in the new Data and Methods section and supported by additional explanations regarding the resampling method.

5. P3 A basic description of data and methods is missing:

   • It would be welcome to provide a short description of the simulations used in this study, that is the control run and MPI-GE. Especially, it should be noted somewhere what RCP is used, and to mention the initialization method that was applied to produce MPI-GE.

   • It should be more clear why the analysis is sometimes applied to MPI-GE or to the preindustrial runs. The resampling methods used in the study should also be discussed.

6. P3L4-5 I would suggest rephrasing "When using a smaller ensemble, sampling uncertainty may be misinterpreted as a forced change in ENSO or a robust difference between two models." to something like: "When using a smaller ensemble, sampling uncertainty may lead to false detection of a forced change in ENSO or a robust difference between two models."

7. P3L8-10 The point that the required ensemble depends on the model (i.e. the magnitude of internal variability) is important and should be discussed further in conclusion.

8. P3L13 "Therefore we differentiate three types of questions that encompass the specific questions that are commonly addressed with a large ensemble and show examples for each type of question" – This sentence needs to be simplified.

9. P3L19-24 I think this section on the resampling problem should rather begin by justifying why one should in the first place resample to estimate the required ensemble size. Then, to describe the different possible resampling approaches in order to justify which one to use in which context (and according to either MPI-GE or the preindustrial runs).

10. P4L3 and P4L12: The choice of resampling without replacement is had hoc and this choice should have been discussed earlier.

11. P4L12-14 "At some point, the 1000 random subsamples are not independent anymore because they share many of the randomly drawn members from the full ensemble." I would highly suggest the authors to compare the number of possible ensembles that can be formed without and with replacement. The second approach offers much more degrees of freedom.

12. Fig. 1: Choose another color for the full envelope (1 member) as it is the same (light blue) as for the 50-member ensemble. Adjust the legend accordingly. A

version of this figure generated by resampling with replacement would add a non-zero uncertainty on the 200-member average.

13. P5L5-6 "For a smaller number of realisations in the full ensemble, the resampling starts to dominate the error convergence earlier than in a much larger ensemble." See general comment on the RP.

14. P5l11013 "The sample size for which the RMSE estimate in a smaller maximum ensemble size starts to diverge from the RMSE estimate based on a larger maximum ensemble size determines the threshold of where resampling substantially affects the error convergence." Here the 50% limit is estimated rather loosely. Comparing versions "with" and "without" replacement of Fig. 2 would give a good indication of where this limit could be. However, I'm not sure this is a very useful result since the alternative approach of resampling with replacement would attenuate the RP, at least for ensemble sizes smaller or equal to 200.

15. Fig. 3:

   - The caption should obviously be re-written and clarified.
   - Results would be more clear by inverting the order of plotting, that is red to light blue from top to bottom.
   - How can a standard deviation have negative values ?

16. P6L1-2 Are the subsamples overlapping or completely independent ? It seems they are overlapping, which might lead to an underestimation of the standard deviation of the distribution due to the serial dependence of the time windows. Generating 50-year periods by randomly resampling individual years could allow to circumvent this issue. The selection of the best approach for this problem should be discussed in the new Data and Methods section.

17. Fig. 4 and 5: Why not using all 200 members with replacement here ? This could allow to get rid of the saturation over the continents. In addition, it would be useful to know exactly over which period these maps are computed.

18. P7L21 "[...] while larger ensemble sizes are affected by resampling and therefore not shown." See general comment on the RP.

19. P7L27-28 "Beyond 50 members, the resampling problem inhibits reliable estimates of the sufficient ensemble size." See general comment on the RP.

20. P11L12-13 "The advantage of this approach, in contrast to the examples for the forced response, is that the required ensemble size can be estimated for any model without needing a large ensemble to be available." Yes – but is this approach (of splitting in overlapping windows) give similar results to a resampling over MPI-GE ? This should be verified by the authors and clarified in the methods section.

21. P11L18 (fig. 8) Same as previous comment about the overlapping windows.

22. p14L9-13 See general comment on the RP.

23. p15l17-18 It would be good to recall some examples from the introduction where other studies have assessed required ensembles for different applications, and compare with the results presented in the current paper.

24. Conclusion: Put important findings in the context of other studies cited in literature. Also discuss that ensemble sizes would likely be different with other models with different magnitude of internal variability.

---

## Referee Comment (RC2) · Anonymous Referee #2 · 10 Feb 2020

This manuscript is investigating the optimal number of members from single-model ensemble. To do so, they are suggesting a conceptual recipe which should provide the optimal number of members. They subdivide their investigation into three sections where they: 1) quantify the forced signal, 2) the internal variability and 3) the change in internal variability in order to provide the optimal number of members for each question using the MPI-Grand Ensemble. The study is showing some interesting results and is worth publishing. However, the writing could be improved (still some internal notes). Since the paper do not really fulfill its promises in a convincing way (providing the size of a large ensemble), the focus of the paper should be rethought. I will therefore suggest accepting the manuscript but only after a major revision. I hope that my comment will help the authors to improve the quality of their paper.

Major comments: Some of the results of this study are interesting and deserve to be published. However, I think the title is not representing the paper, since there is no

concrete conclusion about the number of members, the question remains still an open question which depends on where (regions), what (which variables), who (models) and when (periods), which is already shown in previous study about internal variability. I would suggest changing the whole structure of the paper.

The introduction does not match the rest of the paper. For example, there are three interesting questions at the end of the introduction, but then the paper since to be structured otherwise while suggesting that the recipe for estimating the ensemble size will be followed... It would greatly improve the clarity of the manuscript if the questions were explicitly addressed in the next sections (as subsection). I would suggest transferring this whole discussion of Sect.2 (but removing its main conclusion (see below)) into an Apendix section.

In Sect.2, the authors are investigating at which size the reduction of error is due to the increase of ensemble members and not to the resampling error (or the limits between those two). I fully appreciate the need for such an approach for your studies, however, I do not agree with your conclusion of lines 14 to 16. It may be true for the max ensemble size of 20, but not for the others...It is, at least, highly disputable. I do not see, and therefore not convinced, that the diverging point is ∼50% of the maximum ensemble size. I think that this is the weakest point of the manuscript, but quite important. However, I do not think that this is a deal breaker, since most of the text can me readjust (for example page 7, line 29; page 9 line 9; etc. . .). The following line seems to bring news proofs, but unfortunately I couldn't convince myself otherwise since the text was not clear and accompanied by still some internal notes shielding doubts about the figure (see captions of Fig.3). I would also suggest getting rid of the whole part of page 5 line 17 (or just mention it).

As written, the authors directly proposed a recipe for estimating the ensemble size, which (and I am sorry to say it) look like it is drawn from a hat. I do not understand why (and where) this comes up and why it is presented in that section. As presented, the recipe is stating the obvious and is presented as the center issues of the manuscript,

but is not anyway. I would first specifically answered the tree questions and then maybe proposed a recipe that could be tested in a small paragraph just before the conclusion. In that sense, I think that the manuscript is showing some interesting results, but not fulfilling his promises...

One more general comment, I often had the impression that the solution when choosing the size of the ensemble was to select subsample members of a large ensemble, which for me did not make sense since the whole ensemble should be used (otherwise, why running it?).

Minor comments:

Page 5 line 3-13: This whole paragraph was a bit obscure to me and could be clearer. It needed more details and terms should be explicitly mentioned (and maybe shown on Fig. 2 directly as an example) in the text, such as "the error convergence" in "the resampling start to dominate the error convergence".

Page 7 line 16-20: Those few sentences are quite confusing, could you please add more explanations? In figure 4 a–c, the expected RMSE for each grid point is shown for ensemble sizes of 3, 5, 10, and 50 members. The RMSE is computed as the mean difference between 100 samples (of what of each ensemble size (like in Sect 2, 100 samples of sets of 3,5,10 and 50 members)? If yes, why not have chosen 1000 random samples as in Sect2) and the 100-member mean (which is the whole ensemble, right?). When the ensemble mean is based on just 3 members (so which one? The ensemble-mean of the 100 samples of set of 3 members?), the expected error in the estimated forced response is large over land regions, in particular in the northern hemisphere.

Page 7 line 25-27: ...the acceptable error is 0.1°C... do you mean the number of members needed to restrain the RSME to 0.1°C? If yes, please keep RSME instead of error. Otherwise, please clarify.

The manuscript should have a section explaining the MPI-LA set-up, so the paper can

stand by himself.

Please specify somewhere what is GSAT and Nino3.4

Page 2, line 3-5: I would explicitly mention the term signal-to-noise ratio in that paragraph.

Page 2, line 9-10-11 "If the signal...present-day conditions" I suggest getting rid of that line. I do not like this statement imply that there is enough members to quantify IV, so why would you look only one member. It is irrelevant.

Page 2, line 16: ..of the large regional variability. . .

Page 2 line 16 to 20: I think this is not correctly cited. One the reason that Li and Ilyina (2018) required so many members are most likely due to the week(er) overall forced signals from RCP4.5. As written, it seems that the two studies are comparable (Li and Ilyina (2018) and Steinman et al. (2015)), but their differences should be explicitly mentioned.

Page 2 line 24-28: Please reformulate, not clear. For example, they analyze the polar cortex but concluded about the lower latitude...

Page 2, line 33-34: Could you elaborate a little on that?

Page 5 Figure2: I would change to yellow color for another one...I do not see it well when printed...

---

## Author Comment (AC1) · 7 May 2020

*Thank you for your thoughtful review and suggestions for improving the manuscript. We are happy that you are interested in our results and appreciate your suggestions for improving the manuscript. Please find our replies to your comments below.*

General comments:

In this paper, the authors study the impact of ensemble size on the estimation of different climate statistics using the MPI Grand Ensemble and a pre-industrial control simulation. They analyze the statistical error associated with different quantities as estimated from ensembles of varying sizes, such as the forced response in global surface air temperature, as well as in regional temperature and precipitation. They also assessed the required ensemble size for estimating ENSO variability, linear warming/cooling trends, and changes in internal variability for Arctic sea ice.

Overall, I think this study is highly relevant for guiding users on required ensemble sizes related to different applications, as well as to provide useful insights to climate modellers in the context of the production of upcoming large ensembles. The paper is generally well written and results are original, interesting and worth publishing. However, there are a few sections that would need to be revisited. For instance, I think a short additional section providing a basic description of the "Data and Methods" would make the paper much easier to understand. In addition, I have some concerns about the selected methods, whose details and implications should be discussed in more details. Finally, the conclusions should better put the original findings into a wider context, especially by comparing with other existing studies (as cited in the introduction) that also have estimated required ensemble sizes.

*We will add a short section describing the model and simulations used.*

*However, we would like to keep the description of the method connected to the applications. The primary goal of this study is to develop a method that can be applied to estimate the required ensemble size in any given context. The applications of this method are meant to demonstrate our reasoning for the chosen method and illustrate caveats in the interpretation.*

*In the conclusion section of our revised manuscript, we will discuss our results in the context of the previous studies.*

My main concern about the methodology used in this paper is the exaggerated importance of what the authors call the "resampling problem" (RP). If the aim of this paper is to provide robust estimates of the required ensemble size for different applications (as stated several times in the paper), the importance given to the RP is an obstacle to this goal. The RP is

actually an artifact of the selected strategy of resampling the large ensemble without replacement and has profound impacts on the interpretation of the results. With this approach, the question of "How large does a large ensemble need to be?" becomes highly conditional to the size of the ensemble at hand, especially when 50% (here loosely estimated) of the maximum ensemble size is exceeded. If the author would replace their strategy by resampling WITH replacement, the RP would also become a limitation at some point, but for much larger sample sizes (probably even above than the actual maximum ensemble size of 200 members).

*Thank you for this comment that has stimulated us to rethink how we address the resampling problem, and how we present it in the manuscript.*

*We have realised that the current structure is not ideal. The resampling problem is mentioned very prominently, but too early so that the relevant context is missing. In the revised manuscript, we intend to start with the forced response in GSAT example (fig. 1) and the associated estimate of the required ensemble size (fig. 2). Building on this, we will then integrate the discussion of the resampling problem. We will try to limit this to the extent that is necessary to follow the argument and provide additional information in the supplementary information.*

*Regarding the sampling approach, we made a conscious decision to resample without replacement. The reasoning behind this is that by subsampling for example 5 out of the 200 members, we try to imitate a situation where we only produced 5 members with our model. These could be any 5 out of the 200 members we actually have.*
*In the case where we resample with replacement, a single member could appear more than once in this sample of 5. We think this is unlikely to happen in reality because that would mean that two members produced by a climate model are (nearly) bit-identical despite a different initialisation. By allowing replacement, we would arrive at an arguably too conservative estimate of the required ensemble size.*

*However, we do note that sampling with replacement would be an obvious solution to the problem we raise from a purely statistical perspective. We will therefore include a better reasoning for our choice in the revised manuscript.*
*Our current reasoning for interpreting only up to 50% of the maximum available ensemble size is currently based on an empirical assessment of this threshold. We are working on an additional analytical reasoning to support the choice of this threshold. This will be limited to the effects of resampling without replacement for the mean (i.e. forced response). The 'standard error of the mean' is closely connected to the problem at hand and can be estimated. For higher order moments, this will be less straightforward.*

The previous comment mainly applies to the results based on MPI-GE, but the issue of the resampling strategy also applies to the results based on the pre-industrial control simulation. For this part, the authors do the resampling by generating synthetic members obtained by splitting the pre-industrial control into overlapping segments (e.g. 50 or 100 years). However, three resampling strategies were actually possible, without any explicit mention in the

document: 1) overlapping segments (suffering from the serial dependence of the windows), 2) non-overlapping segments (leading to only 20 members from the 2000-year time series), and 3) random year selection to generate synthetic segments (either with or without replacement). Implications and interpretation of these possible approaches should be discussed in order to support the decision of selecting which one is better to apply in which context.

*Yes, the resampling does indeed have implications for the analysis based on the pre-industrial control simulation. We will elaborate on this in the revised manuscript and provide a reasoning for the strategy we used.*

Specific comments:

1. p1l7-8 "First, we determine how much of an available ensemble size is interpretable without a substantial impact of resampling ensemble members" The RP is a limitation of the current approach and could be attenuated by changing the resampling approach. I don't think this issue should be mentioned in the abstract, and other similar comments in the paper should be revisited according to the above general comment on RP.

*As outlined above, we will make substantial changes to the treatment and presentation of the resampling problem in the manuscript.*

*However, we do think that the resampling problem is an important caveat that needs to be considered when determining the required ensemble size. Previous studies have concluded that X of N ensemble members are sufficient to detect a signal, with X/N being around 0.6-0.8. Therefore, we felt that this potential caveat should be highlighted in the abstract.*

2. P2L13: "to to"

*Noted, thank you.*

3. P2L22-24: I think the reference to Pausata et al. (2015) is not correct. Maybe another paper from the same author is cited ?

*Yes, this is indeed the wrong reference. We will change this to the correct reference: Pausata, F. S. R., Grini, A., Caballero, R., Hannachi, A. & Seland, Ø. High-latitude volcanic eruptions in the Norwegian Earth System Model: the effect of different initial conditions and of the ensemble size. Tellus B: Chemical and Physical Meteorology 67, 26728–17 (2015).*

4. P1L24 "make use of a model's pre-industrial control run where possible." This is not that clear in the paper why sometimes we use MPI-GE and otherwise the preindustrial run. This should be clarified in the new Data and Methods section and supported by additional explanations regarding the resampling method.

*Agreed, we will make sure to explain in more detail under which conditions the control run can be used, and how this can be done in practice.*

5. P3 A basic description of data and methods is missing:
   ● It would be welcome to provide a short description of the simulations used in this study, that is the control run and MPI-GE. Especially, it should be noted somewhere what RCP is used, and to mention the initialization method that was applied to produce MPI-GE.

*We will add a short section explaining the design of the MPI-GE and the runs used in this study. (pre-industrial control, historical, and 1% $CO_2$)*

   ● It should be more clear why the analysis is sometimes applied to MPI-GE or to the preindustrial runs. The resampling methods used in the study should also be discussed.

*Our objective is to use the preindustrial control run whenever possible because this simulation is readily available for every CMIP5/6 model, while a large ensemble is not. However, some of the applications require a different type of simulation where the external forcing is changing over time (increasing CO2, volcanic eruptions). In the revised manuscript, we will make this choice more clear. Our recommendation is to use a preindustrial control simulation when possible. As noted in response to comment 4, we will include more detailed recommendations for how the pre-industrial control simulation can be used for specific questions.*

6. P3L4-5 I would suggest rephrasing "When using a smaller ensemble, sampling uncertainty may be misinterpreted as a forced change in ENSO or a robust difference between two models." to something like: "When using a smaller ensemble, sampling uncertainty may lead to false detection of a forced change in ENSO or a robust difference between two models."

*Thank you, we will follow your suggestion.*

7. P3L8-10 The point that the required ensemble depends on the model (i.e. the magnitude of internal variability) is important and should be discussed further in conclusion.

*Thank you for this suggestion. It seems that this point was not clear enough and we will make sure to emphasise this more. Our motivation for introducing a method rather than recommended ensemble sizes is based on this point: analysis of a different model might result in a different required ensemble size. Therefore we suggest that this analysis is repeated with every model before using it, rather than assuming that the required ensemble size derived from the MPI-ESM in this study is valid for all models.*

8. P3L13 "Therefore we differentiate three types of questions that encompass the specific questions that are commonly addressed with a large ensemble and show examples for each type of question" – This sentence needs to be simplified.

*Agreed. If this sentence is still in the manuscript after rewriting, we will make sure to simplify it.*

9. P3L19-24 I think this section on the resampling problem should rather begin by justifying why one should in the first place resample to estimate the required ensemble size. Then, to describe the different possible resampling approaches in order to justify which one to use in which context (and according to either MPI- GE or the preindustrial runs).

*As mentioned above, we intend to restructure the sections to provide more background before mentioning the resampling problem. The updated structure will be:*

- *Introduction*
- *Model description*
- *The basic approach for estimating the required ensemble size (forced signal in GSAT and regional temperature and precipitation)*
    - *the resampling problem*
- *recipe*
- *applying the recipe to various typical problems*
- *…*

10. P4L3 and P4L12: The choice of resampling without replacement is had hoc and this choice should have been discussed earlier.

*Yes, as stated above we will justify our resampling approach and discuss the alternative approach with replacement.*

11. P4L12-14 "At some point, the 1000 random subsamples are not independent anymore because they share many of the randomly drawn members from the full ensemble." I would highly suggest the authors to compare the number of possible ensembles that can be formed without and with replacement. The second approach offers much more degrees of freedom.

*As described in the responses to comments 4 and 5, we intend to extend the discussion of the use of the control run, including the sampling strategy for different types of question.*

12. Fig. 1: Choose another color for the full envelope (1 member) as it is the same (light blue) as for the 50-member ensemble. Adjust the legend accordingly. A version of this figure generated by resampling with replacement would add a non-zero uncertainty on the 200-member average.

*We will change the color as suggested. Yes, resampling with replacement indeed adds a non-zero uncertainty for the 200-member average and increases the uncertainty for most other averages. As described above, we believe that this estimate would be too conservative. We choose the approach that provides a less conservative uncertainty estimate, but introduces the resampling problem. We will discuss this in more detail in the revised manuscript.*

13. P5L5-6 "For a smaller number of realisations in the full ensemble, the resampling starts to dominate the error convergence earlier than in a much larger ensemble." See general comment on the RP.

*Noted.*

14. P5l11013 "The sample size for which the RMSE estimate in a smaller maximum ensemble size starts to diverge from the RMSE estimate based on a larger maximum ensemble size determines the threshold of where resampling substantially affects the error convergence." Here the 50% limit is estimated rather loosely. Comparing versions "with" and "without" replacement of Fig. 2 would give a good indication of where this limit could be. However, I'm not sure this is a very useful result since the alternative approach of resampling with replacement would attenuate the RP, at least for ensemble sizes smaller or equal to 200.

*We will support the empirically estimated 50% threshold with a more rigorous derivation of the sample size at which resampling affects the conclusion. As described above, this is more straightforward for the mean than for higher order moments, which is why we relied on the empirical approach in the submitted version of this manuscript.*

15. Fig. 3:
    • The caption should obviously be re-written and clarified.
    • Results would be more clear by inverting the order of plotting, that is red to light blue from top to bottom.
    • How can a standard deviation have negative values ?

*Apologies for including an old caption in the submitted manuscript. The figure was updated, but not the caption.The caption should read:*

> *PDF of ensemble-averaged Niño3.4 standard deviations possible in the MPI-GE pre-industrial control simulation for subsampling ensembles ranging from 50 to 1000 members (shown as different colors) for smaller ensemble sizes. Each PDF is shown relative to the corresponding ensemble mean value. We use the last 1000 years of the 2000 year control run to calculate the ranges. The Niño3.4 standard deviation is calculated over 50 year periods. The PDFs are created by resampling the control simulation 1000 times. For each PDF the entirety of the 1000 years are used (i.e. the blue 500 member pdf is the mean of 2 500 members PDFs).*

*We will invert the order of plotting as suggested.*

*The standard deviation is relative to the mean value, this is now clarified in the caption.*

16. P6L1-2 Are the subsamples overlapping or completely independent ? It seems they are overlapping, which might lead to an underestimation of the standard deviation of the distribution due to the serial dependence of the time windows. Generating 50-year periods by randomly resampling individual years could allow to circumvent this issue. The selection of the best approach for this problem should be discussed in the new Data and Methods section.

*The subsamples are overlapping. We will explain this in more detail in the revised manuscript. In the case where we quantify ENSO variability, random resampling would not be representative of real ensemble members because ENSO has a timescale larger than 1 year. We selected consecutive years to retain the temporal characteristics of ENSO.*

17. Fig. 4 and 5: Why not using all 200 members with replacement here ? This could allow to get rid of the saturation over the continents. In addition, it would be useful to know exactly over which period these maps are computed.

*We did repeat the analysis with all 200 members (figures below to replace figure 4 and 5). The period is the full length of the historical simulations (1850–2005). We will make this more clear in the revised text. This analysis is an extension of the analysis in figure 1 and 2. For each grid point, we show the expected RMSE at a specific ensemble size, which is equivalent to the value of the solid black line in figure 2 for that ensemble size (computed for a grid point instead of globally).*

[Figure]

18. P7L21 "[. . . ] while larger ensemble sizes are affected by resampling and therefore not shown." See general comment on the RP.

*Noted.*

19. P7L27-28 "Beyond 50 members, the resampling problem inhibits reliable estimates of the sufficient ensemble size." See general comment on the RP.

*Noted.*

20. P11L12-13 "The advantage of this approach, in contrast to the examples for the forced response, is that the required ensemble size can be estimated for any model without needing a large ensemble to be available." Yes – but is this approach (of splitting in overlapping windows) give similar results to a resampling over MPI-GE ? This should be verified by the authors and clarified in the methods section.

*Yes, sampling over several years in the control run and sampling over members in the MPI-GE does provide results, under the condition that the forcing in the MPI-GE has not changed the distribution. We will clarify this in the revised manuscript.*

21. P11L18 (fig. 8) Same as previous comment about the overlapping windows.

*Noted.*

22. p14L9-13 See general comment on the RP.

*Noted.*

23. p15l17-18 It would be good to recall some examples from the introduction where other studies have assessed required ensembles for different applications, and compare with the results presented in the current paper.

*We will make sure that the discussion revisits questions raised in the introduction. However, we do not want to reproduce the analysis of previous studies in detail because we do not want to put too much focus on the actual numbers that we find in this study. Our main objective is to present a generic method that can be used to determine the required ensemble size, explain caveats, and how it can be applied in practice.*

24. Conclusion: Put important findings in the context of other studies cited in literature. Also discuss that ensemble sizes would likely be different with other models with different magnitude of internal variability.

*We agree that the results are possibly highly model dependent. This is an important point and we will emphasise this in the revised conclusion.*

---

## Author Comment (AC2) · 7 May 2020

This manuscript is investigating the optimal number of members from single-model ensemble. To do so, they are suggesting a conceptual recipe which should provide the optimal number of members. They subdivide their investigation into three sections where they: 1) quantify the forced signal, 2) the internal variability and 3) the change in internal variability in order to provide the optimal number of members for each question using the MPI-Grand Ensemble. The study is showing some interesting results and is worth publishing. However, the writing could be improved (still some internal notes). Since the paper do not really fulfill its promises in a convincing way (providing the size of a large ensemble), the focus of the paper should be rethought. I will therefore suggest accepting the manuscript but only after a major revision. I hope that my comment will help the authors to improve the quality of their paper.

*Thank you for your thoughtful review and suggestions for improving the manuscript. Our main objective is to suggest a conceptual recipe to estimate the required ensemble size, explain the reasoning for using the recipe, and discuss possible caveats in the interpretation of the results. We do provide the required ensemble sizes for several applications in the MPI-GE. These applications are meant to demonstrate how the method can be applied. The required ensemble sizes we find are likely dependent on the model used (its magnitude of internal variability and the relative magnitude of the investigated signal). We acknowledge that we need to ascertain that these objectives are clearly stated in the revised manuscript, so that our results meet the reader's expectations.*

*We apologise for not removing the internal notes in the caption of figure 3.*

Major comments: Some of the results of this study are interesting and deserve to be published. However, I think the title is not representing the paper, since there is no concrete conclusion about the number of members, the question remains still an open question which depends on where (regions), what (which variables), who (models) and when (periods), which is already shown in previous study about internal variability. I would suggest changing the whole structure of the paper.

*Our intention was not to provide a conclusion about the numbers of ensemble members needed, because such a number does indeed depend on the specific question asked (region, variable) and the climate model used. Instead, we propose a generic method that can be used to estimate the required ensemble size for any given question and any climate model. The method can either be applied to an existing large ensemble to test if it is the right tool for the question at hand, but it can also be applied to a pre-industrial control run to estimate the required ensemble size before running a new large ensemble. In the revised manuscript, we will elaborate more on the option to use the pre-industrial control run of a model. (see our replies to reviewer 1 comments 4,5,7,16,20)*

The introduction does not match the rest of the paper. For example, there are three interesting questions at the end of the introduction, but then the paper since to be structured otherwise while suggesting that the recipe for estimating the ensemble size will be followed... It would greatly improve the clarity of the manuscript if the questions were explicitly addressed in the next sections (as subsection). I would suggest transferring this whole discussion of Sect.2 (but removing its main conclusion (see below)) into an Apendix section.

*We have realised that the resampling problem is mentioned too early and without the appropriate context. We will restructure the sections to provide more background before mentioning the resampling problem. The updated structure will be:*

- *Introduction*
- *Model description*
- *The basic approach for estimating the required ensemble size (forced signal in GSAT and regional temperature and precipitation)*
    - *the resampling problem*
- *recipe*
- *applying the recipe to various typical problems*
- *…*

*The applications of the recipe follow the three questions outlined in the introduction. We believe that the updated structure will be much easier to follow.*

*The applications follow the three questions in the introduction (currently sections 4.1 to 4.3)*

*1)response to external forcing: GSAT, regional temperature and precipitation, linear warming trend, cooling after volcanic eruption*
*2) quantify internal variability: ENSO and temperature variability over land*
*3) identify a forced change in variability: Arctic sea ice area*

*The updated structure of the revised manuscript will make it easier to link the examples to the three questions. We will also add a paragraph to the conclusions where we will link the examples to the respective question.*

In Sect.2, the authors are investigating at which size the reduction of error is due to the increase of ensemble members and not to the resampling error (or the limits between those two). I fully appreciate the need for such an approach for your studies, however, I do not agree with your conclusion of lines 14 to 16. It may be true for the max ensemble size of 20,

but not for the others...It is, at least, highly disputable. I do not see, and therefore not convinced, that the diverging point is ~50% of the maximum ensemble size. I think that this is the weakest point of the manuscript, but quite important. However, I do not think that this is a deal breaker, since most of the text can me readjust (for example page 7, line 29; page 9 line 9; etc. . .). The following line seems to bring news proofs, but unfortunately I couldn't convince myself otherwise since the text was not clear and accompanied by still some internal notes shielding doubts about the figure (see captions of Fig.3). I would also suggest getting rid of the whole part of page 5 line 17 (or just mention it).

*The approach we take to resampling will be updated and explained in more detail in the revised manuscript to take the suggestions by reviewer 1 into account. We will also include more theoretical background to our choice of 50%, which is currently based on an empirical approach.*

*We apologise for the internal note in the caption of figure 3. The figure itself has been updated, but we did not update the figure caption. The figure caption should read:*

> *PDF of ensemble-averaged Niño3.4 standard deviations possible in the MPI-GE pre-industrial control simulation for subsampling ensembles ranging from 50 to 1000 members (shown as different colors) for smaller ensemble sizes. Each PDF is shown relative to the corresponding ensemble mean value. We use the last 1000 years of the 2000 year control run to calculate the ranges. The Niño3.4 standard deviation is calculated over 50 year periods. The PDFs are created by resampling the control simulation 1000 times. For each PDF the entirety of the 1000 years are used (i.e. the blue 500 member pdf is the mean of 2 500 members PDFs).*

As written, the authors directly proposed a recipe for estimating the ensemble size, which (and I am sorry to say it) look like it is drawn from a hat. I do not understand why (and where) this comes up and why it is presented in that section. As presented, the recipe is stating the obvious and is presented as the center issues of the manuscript, but is not anyway. I would first specifically answered the tree questions and then maybe proposed a recipe that could be tested in a small paragraph just before the conclusion. In that sense, I think that the manuscript is showing some interesting results, but not fulfilling his promises…

*We will solve this problem by the updated structure (see comment above). We use the forced response in historical GSAT as an example to illustrate how the question from the title can be approached and how resampling can become an issue. We will then use the examples of regional temperature and precipitation to demonstrate how different variables, regions, and acceptable errors influence the required ensemble size.*

*This will then be followed by the recipe and all remaining examples, all of which illustrate how the recipe can be applied to the three types of question mentioned in the introduction.*

One more general comment, I often had the impression that the solution when choosing the size of the ensemble was to select subsample members of a large ensemble, which for me did not make sense since the whole ensemble should be used (otherwise, why running it?).

*Here we take advantage of an existing very large 200-member ensemble. The advantage of using this ensemble is that the full ensemble is likely very close to the truth for many applications. In the case of GSAT, the 200-member mean provides a good reference for the true forced response in this model. We can then ask: how large is the error when using the ensemble mean of a smaller ensemble to estimate the model's forced response? We answer this question by subsampling the full ensemble.*

*When using the MPI-GE, one would certainly use all available members. In the context of this study, the 200 members from the MPI-GE allow us to explore how well our recipe works for other typical ensemble sizes of large ensembles (e.g. figure 2). Finally, we demonstrate how a pre-industrial control simulation can be used to estimate the required ensemble size for a given model and question. This approach can be used to determine which models from the CMIP5 or CMIP6 archive provide a sufficient number of realisations, or it can be used to determine the ensemble size required for a variety of questions before running a new large ensemble.*

Minor comments:

Page 5 line 3-13: This whole paragraph was a bit obscure to me and could be clearer. It needed more details and terms should be explicitly mentioned (and maybe shown on Fig. 2 directly as an example) in the text, such as "the error convergence" in "the resampling start to dominate the error convergence".

*Thank you, we have realised that the necessary context for this paragraph is only mentioned later in the manuscript. We will improve this when restructuring the manuscript.*

Page 7 line 16-20: Those few sentences are quite confusing, could you please add more explanations? In figure 4 a–c, the expected RMSE for each grid point is shown for ensemble sizes of 3, 5, 10, and 50 members. The RMSE is computed as the mean difference between 100 samples (of what of each ensemble size (like in Sect 2, 100 samples of sets of 3,5,10 and 50 members)? If yes, why not have chosen 1000 random samples as in Sect2) and the 100-member mean (which is the whole ensemble, right?). When the ensemble mean is based on just 3 members (so which one? The ensemble- mean of the 100 samples of set of 3 members?), the expected error in the estimated forced response is large over land regions, in particular in the northern hemisphere.

*We will extend the explanation. The RMSE for a grid point and ensemble size (e.g. figure 4a for 3 members) essentially contains the same information as the solid black line in figure 2 at an ensemble size of 3 (of course after recomputing this for the regional instead of global temperature). Note that the maps in figures 4 and 5 only represent the expected RMSE and not the uncertainty interval (shading in figure 2).*

*The sample size of 100 instead of 1000 was selected because this analysis is computationally expensive. We will consider reducing the sample size in figure 2 to be more consistent*

*Note that we updated figures 4 and 5 to use all 200 members instead of 100. (see response to reviewer 1, comment 17)*

Page 7 line 25-27: ...the acceptable error is 0.1◦C... do you mean the number of members needed to restrain the RSME to 0.1◦C? If yes, please keep RSME instead of error. Otherwise, please clarify.

*Yes, acceptable error refers to RMSE in this context. We will clarify this in the revised manuscript.*

The manuscript should have a section explaining the MPI-LA set-up, so the paper can stand by himself.

*We will add a section describing the model and experiments used after the introduction.*

Please specify somewhere what is GSAT and Nino3.4

*Thank you, we will add the definitions.*

Page 2, line 3-5: I would explicitly mention the term signal-to-noise ratio in that para- graph.

*Thank you for this suggestion. In this paragraph, we want to introduce the concept of averaging over many ensemble members to eliminate the noise from internal variability. This approach is not directly evaluating the ratio between the signal from the forced response to the noise from internal variability.*

Page 2, line 9-10-11 "If the signal...present-day conditions" I suggest getting rid of that line. I do not like this statement imply that there is enough members to quantify IV, so why would you look only one member. It is irrelevant.

*Here, we should have explicitly used the term signal-to-noise. We look at the question: how many members do we need to be certain that a signal exists. In the case where a single trajectory clearly emerges from the noise of internal variability, the presence of a signal can be detected in a single realisation. For example, a single RCP8.5 realisation is clearly sufficient to conclude that the end of the 21st century is warmer than pre-industrial conditions. In the introduction, we wanted to mention that not all applications require a large ensemble.*

Page 2, line 16: ..of the large regional variability. . .

*Thank you, this will be changed.*

Page 2 line 16 to 20: I think this is not correctly cited. One the reason that Li and Ilyina (2018) required so many members are most likely due to the week(er) overall forced signals

from RCP4.5. As written, it seems that the two studies are comparable (Li and Ilyina (2018) and Steinman et al. (2015)), but their differences should be explicitly mentioned.

*Thank you, we will extend the description of these papers. Li and Ilyina investigate carbon uptake in the southern ocean. In this case, the signal-to-noise ratio is small because of the large variability in the southern ocean. Steinmann et al. investigate a region and quantity that is less variable and has a larger forced signal, therefore the signal-to-noise ratio is large and a small number of ensemble members is required. The only similarity between the two studies is that they try to identify a forced change. We choose these examples to illustrate the large differences in ensemble size requirements when the investigated quantity and region are different.*

Page 2 line 24-28: Please reformulate, not clear. For example, they analyze the polar cortex but concluded about the lower latitude…

*We will extend the description.*

Page 2, line 33-34: Could you elaborate a little on that?

*We will extend this paragraph. Here, we introduce a common strategy: instead of using a large ensemble, a long pre-industrial control run with no change in the external forcing is used to quantify internal variability. This estimate of internal variability can then be used to quantify the uncertainty due to internal variability in simulations where the external forcing is changing. The underlying assumption is that internal variability does not change when the external forcing is changing. While this is true for some quantities, it does not hold for other quantities such as the Arctic sea ice area as we show in section 4.3.*

Page 5 Figure2: I would change to yellow color for another one...I do not see it well when printed...

*Thank you for the suggestion. We will revisit the choice of colors for this figure.*

---

## Author Response (AR1)

**Received and published: 22 January 2020**

Thank you for your thoughtful review and suggestions for improving the manuscript. We are happy that you are interested in our results and appreciate your suggestions for improving the manuscript. Please find our replies to your comments below.

**General comments:**

In this paper, the authors study the impact of ensemble size on the estimation of different climate statistics using the MPI Grand Ensemble and a pre-industrial control simulation. They analyze the statistical error associated with different quantities as estimated from ensembles of varying sizes, such as the forced response in global surface air temperature, as well as in regional temperature and precipitation. They also assessed the required ensemble size for estimating ENSO variability, linear warming/cooling trends, and changes in internal variability for Arctic sea ice.

Overall, I think this study is highly relevant for guiding users on required ensemble sizes related to different applications, as well as to provide useful insights to climate modellers in the context of the production of upcoming large ensembles. The paper is generally well written and results are original, interesting and worth publishing. However, there are a few sections that would need to be revisited. For instance, I think a short additional section providing a basic description of the "Data and Methods" would make the paper much easier to understand. In addition, I have some concerns about the selected methods, whose details and implications should be discussed in more details. Finally, the conclusions should better put the original findings into a wider context, especially by comparing with other existing studies (as cited in the introduction) that also have estimated required ensemble sizes.

We added a short section describing the model and simulations used.

However, we would like to keep the description of the method connected to the applications. The primary goal of this study is to develop a method that can be applied to estimate the required ensemble size in any given context. The applications of this method are meant to demonstrate our reasoning for the chosen method and illustrate caveats in the interpretation.

We have updated the structure of the paper to introduce the method and the generalised recipe in section 3.

We have added a short paragraph to the conclusions section pointing out similarities to previous studies. However, the example applications in this study are not identical to the applications from previous work. Furthermore, model differences could contribute to different ensemble size requirements. Therefore, we use this section to emphasise that there is no

ensemble size that is sufficient for every model or application, but encourage our readers to estimate the required ensemble specifically for the combination of application and model(s) used.

My main concern about the methodology used in this paper is the exaggerated importance of what the authors call the "resampling problem" (RP). If the aim of this paper is to provide robust estimates of the required ensemble size for different applications (as stated several times in the paper), the importance given to the RP is an obstacle to this goal. The RP is actually an artifact of the selected strategy of resampling the large ensemble without replacement and has profound impacts on the interpretation of the results. With this approach, the question of "How large does a large ensemble need to be?" becomes highly conditional to the size of the ensemble at hand, especially when 50% (here loosely estimated) of the maximum ensemble size is exceeded. If the author would replace their strategy by resampling WITH replacement, the RP would also become a limitation at some point, but for much larger sample sizes (probably even above than the actual maximum ensemble size of 200 members).

Thank you for this comment that has stimulated us to rethink how we address the resampling problem, and how we present it in the manuscript.

We have realised that the current structure is not ideal. The resampling problem is mentioned very prominently, but too early so that the relevant context is missing.

In the revised manuscript, we have reduced the complexity of the GSAT example to focus on the steps to estimate the required ensemble size for this application. The 'recipe' is integrated in this section and directly builds on the GSAT example. The resampling problem is mentioned in a short paragraph. The detailed discussion of issues related to sampling, such as resampling and sampling with or without replacement, has been moved to a new appendix.

Regarding the sampling approach, we made a conscious decision to resample without replacement. The reasoning behind this is that by subsampling for example 5 out of the 200 members, we try to imitate a situation where we only produced 5 members with our model. These could be any 5 out of the 200 members we actually have.

In the case where we resample with replacement, a single member could appear more than once in this sample of 5. We think this is unlikely to happen in reality because that would mean that two members produced by a climate model are (nearly) bit-identical despite a different initialisation. By allowing replacement, we would arrive at an arguably too conservative estimate of the required ensemble size. This can also be seen in the new figure *A*1:

**Figure A1.** Sampling with or without replacement affects the error estimate and therefore the estimate for the required ensemble size. The black line shows the mean RMSE for GSAT for ensemble sizes from 2 to 200. The reference is the 200-member mean from figure 1 and the RMSE is computed for all 1000 samples. The shaded area shows the range of RMSE values for individual samples, the solid line shows the mean RMSE. The red line and shading show the RMSE for ensemble sizes from 2 to 200, but samples are generated by allowing sampling with replacement.

However, we do note that sampling with replacement would be an obvious solution to the problem we raise from a purely statistical perspective. We have therefore explained our choice to sample without replacement in appendix section A1.

Our reasoning for interpreting only up to 50% of the maximum available ensemble size is based on an empirical assessment of this threshold. We have considered an analytical derivation but concluded that this is too complex for the most applications in this study beyond the trivial case where the sampling uncertainty scales with 1/sqrt(n), which is only the case when estimating the mean of a sample generated by a stationary process (e.g. the mean of a pre-industrial control simulation). For higher order moments like the standard deviation or more complex error metrics applied here such as the RMSE, trends, or differences between time periods, estimating the theoretical sampling uncertainty is more complex. Our objective here is to introduce a simple framework that can be modified for a wide range of applications and that can easily be applied.

The previous comment mainly applies to the results based on MPI-GE, but the issue of the resampling strategy also applies to the results based on the pre-industrial control simulation. For this part, the authors do the resampling by generating synthetic members obtained by splitting the pre-industrial control into overlapping segments (e.g. 50 or 100 years). However, three resampling strategies were actually possible, without any explicit mention in the document: 1) overlapping segments (suffering from the serial dependence of the windows), 2) non-overlapping segments (leading to only 20 members from the 2000-year time series), and 3) random year selection to generate synthetic segments (either with or without replacement). Implications and interpretation of these possible approaches should be

discussed in order to support the decision of selecting which one is better to apply in which context.

Yes, the resampling does indeed have implications for the analysis based on the pre-industrial control simulation. We have added a subsection within 4.2 to explain our sampling choice and alternative options and hope this will provide additional value to our readers.

Specific comments:

 p1I7-8 "First, we determine how much of an available ensemble size is interpretable without a substantial impact of resampling ensemble members" The RP is a limitation of the current approach and could be attenuated by changing the resampling approach. I don't think this issue should be mentioned in the abstract, and other similar comments in the paper should be revisited according to the above general comment on RP.

We have removed the RP from the abstract and restructured the paper to put less emphasis on the RP.

We do think that the resampling problem is an important caveat that needs to be considered when determining the required ensemble size. Previous studies have concluded that X of N ensemble members are sufficient to detect a signal, with X/N being around 0.6-0.8.

2. P2L13: "to to"

**Noted, thank you.**

3. P2L22-24: I think the reference to Pausata et al. (2015) is not correct. Maybe another paper from the same author is cited ?

Yes, this is indeed the wrong reference. We changed this to the correct reference: Pausata, F. S. R., Grini, A., Caballero, R., Hannachi, A. & Seland, Ø. High-latitude volcanic eruptions in the Norwegian Earth System Model: the effect of different initial conditions and of the ensemble size. Tellus B: Chemical and Physical Meteorology 67, 26728–17 (2015).

4. P1L24 "make use of a model's pre-industrial control run where possible." This is not that clear in the paper why sometimes we use MPI-GE and otherwise the preindustrial run. This should be clarified in the new Data and Methods section and supported by additional explanations regarding the resampling method.

We have added an additional subsection within 4.2 to discuss the motivation for sampling from the pre-industrial control run and different sampling approaches.

- 5. P3 A basic description of data and methods is missing:
  - It would be welcome to provide a short description of the simulations used in this study, that is the control run and MPI-GE. Especially, it should be noted

somewhere what RCP is used, and to mention the initialization method that was applied to produce MPI-GE.

We added a short model section explaining the design of the MPI-GE and the runs used in this study. (pre-industrial control, CMIP5 historical, and  $1\% CO_2$ ).

The method is central to this study, but we think that the approach is easier to understand when introduced with an example. This is what we do in the revised section 3.

• It should be more clear why the analysis is sometimes applied to MPI-GE or to the preindustrial runs. The resampling methods used in the study should also be discussed.

Our objective is to use the preindustrial control run whenever possible because this simulation is readily available for every CMIP5/6 model, while a large ensemble is not. However, some of the applications require a different type of simulation where the external forcing is changing over time (increasing CO2, volcanic eruptions). In the revised manuscript, we added a new subsection 4.2.1 to discuss sampling in the pre-industrial control simulation.

6. P3L4-5 I would suggest rephrasing "When using a smaller ensemble, sampling uncertainty may be misinterpreted as a forced change in ENSO or a robust difference between two models." to something like: "When using a smaller ensemble, sampling uncertainty may lead to false detection of a forced change in ENSO or a robust difference between two models."

**Thank you, we followed your suggestion.**

7. P3L8-10 The point that the required ensemble depends on the model (i.e. the magnitude of internal variability) is important and should be discussed further in conclusion.

Thank you for this suggestion. It seems that this point was not clear enough. The final two paragraphs in the conclusions now address this in more detail.

8. P3L13 "Therefore we differentiate three types of questions that encompass the specific questions that are commonly addressed with a large ensemble and show examples for each type of question" – This sentence needs to be simplified.

**Agreed. We changed this to: "Therefore we differentiate three types of questions that represent questions typically addressed with large ensembles:"**

 P3L19-24 I think this section on the resampling problem should rather begin by justifying why one should in the first place resample to estimate the required ensemble size. Then, to describe the different possible resampling approaches in order to justify which one to use in which context (and according to either MPI- GE or the preindustrial runs). We moved most of the material on resampling to the appendix and start with a simple example to introduce and explain our method. The resampling is only briefly mentioned as a caveat in the main text.

10. P4L3 and P4L12: The choice of resampling without replacement is had hoc and this choice should have been discussed earlier.

**Resampling with and without replacement is now discussed in appendix A1.**

11. P4L12-14 "At some point, the 1000 random subsamples are not independent anymore because they share many of the randomly drawn members from the full ensemble." I would highly suggest the authors to compare the number of possible ensembles that can be formed without and with replacement. The second approach offers much more degrees of freedom.

It is true that sampling with replacement offers more degrees of freedom. However, this also produces synthetic ensembles that would be treated with suspicion when encountered in an existing large ensemble: an ensemble that contains two (or more) completely identical realisations would raise doubts about the correct initialisation rather than being treated as an ensemble containing fully independent members. We also discuss this in appendix A1.

12. Fig. 1: Choose another color for the full envelope (1 member) as it is the same (light blue) as for the 50-member ensemble. Adjust the legend accordingly. A version of this figure generated by resampling with replacement would add a non-zero uncertainty on the 200-member average.

We changed the color as suggested. A version of figure 2 where we compare sampling with and without replacement is now shown and discussed in appendix A1.

13. P5L5-6 "For a smaller number of realisations in the full ensemble, the resampling starts to dominate the error convergence earlier than in a much larger ensemble." See general comment on the RP.

**Noted. We have made substantial changes to the structure in response to the general comment on the RP.**

14. P5I11013 "The sample size for which the RMSE estimate in a smaller maximum ensemble size starts to diverge from the RMSE estimate based on a larger maximum ensemble size determines the threshold of where resampling substantially affects the error convergence." Here the 50% limit is estimated rather loosely. Comparing versions "with" and "without" replacement of Fig. 2 would give a good indication of where this limit could be. However, I'm not sure this is a very useful result since the alternative approach of resampling with replacement would attenuate the RP, at least for ensemble sizes smaller or equal to 200.

We have moved this discussion to appendix A. 50% is not meant as a strict limit, but as a reminder that ensemble sizes around and beyond this point should be interpreted more carefully.

Our reason for sampling without replacement is explained in A1. (also see response to comment 11)

15. Fig. 3:

• The caption should obviously be re-written and clarified.

• Results would be more clear by inverting the order of plotting, that is red to light blue from top to bottom.

• How can a standard deviation have negative values ?

Apologies for including an old caption in the submitted manuscript. The figure was updated, but not the caption. The caption now reads:

Figure A3. PDF of ensemble-averaged Niño3.4 standard deviations possible in the MPI-GE pre-industrial control simulation for subsampling ensembles ranging from 50 to 1000 members (shown as different colors) for smaller ensemble sizes. Each PDF is shown relative to the corresponding ensemble mean value. We use the last 1000 years of the 2000 year control run to calculate the ranges. The Niño3.4 standard deviation is calculated over 50 year periods. The PDFs are created by resampling the control simulation 1000 times. For each PDF the entirety of the 1000 years are used (i.e. the blue 500 member pdf is the mean of 2 500 members PDFs).

We updated the colors to be consistent with figure A2.

The standard deviation is relative to the mean value, this is now clarified in the caption.

16. P6L1-2 Are the subsamples overlapping or completely independent ? It seems they are overlapping, which might lead to an underestimation of the standard deviation of the distribution due to the serial dependence of the time windows. Generating 50-year periods by randomly resampling individual years could allow to circumvent this issue. The selection of the best approach for this problem should be discussed in the new Data and Methods section.

The subsamples are overlapping. We explain this in more detail in the revised manuscript (also section 4.2.1). In the case where we quantify ENSO variability, random resampling would not be representative of real ensemble members because ENSO has a timescale longer than 1 year. We selected consecutive years to retain the temporal characteristics of ENSO.

17. Fig. 4 and 5: Why not using all 200 members with replacement here ? This could allow to get rid of the saturation over the continents. In addition, it would be useful to know exactly over which period these maps are computed.

We did repeat the analysis with all 200 members (figures below replace figure 4 and 5, now 3 and 4). The period is the full length of the historical simulations (1850–2005). This analysis

is an extension of the analysis in figure 1 and 2. For each grid point, we show the expected RMSE at a specific ensemble size, which is equivalent to the value of the solid black line in figure 2 for that ensemble size (computed for a grid point instead of globally).